# Kaposi's sarcoma-associated herpesvirus (KSHV) utilizes the NDP52/CALCOCO2 selective autophagy receptor to disassemble processing bodies

Carolyn-Ann Robinson[1,2,3,4☯], Gillian K. Singh[1☯], Mariel Kleer[2,3,4☯], Thalia Katsademas[2,3,4], Elizabeth L. Castle[1,3], Bre Q. Boudreau[1], Jennifer A. Corcoran[1,2,3,4]*

1 Department of Microbiology & Immunology, Dalhousie University, Halifax, Canada, 2 Microbiology, Immunology & Infectious Diseases Department, University of Calgary, Calgary, Canada, 3 Charbonneau Cancer Research Institute, University of Calgary, Calgary, Canada, 4 Snyder Institute for Chronic Diseases, University of Calgary, Canada

☯ These authors contributed equally to this work.
* jennifer.corcoran@ucalgary.ca

**Data Availability Statement:** All relevant data are within the manuscript and its Supporting Information files.

## Abstract

Kaposi's sarcoma-associated herpesvirus (KSHV) causes the inflammatory and angiogenic endothelial cell neoplasm, Kaposi's sarcoma (KS). We previously demonstrated that the KSHV Kaposin B (KapB) protein promotes inflammation via the disassembly of cytoplasmic ribonucleoprotein granules called processing bodies (PBs). PBs modify gene expression by silencing or degrading labile messenger RNAs (mRNAs), including many transcripts that encode inflammatory or angiogenic proteins associated with KS disease. Although our work implicated PB disassembly as one of the causes of inflammation during KSHV infection, the precise mechanism used by KapB to elicit PB disassembly was unclear. Here we reveal a new connection between the degradative process of autophagy and PB disassembly. We show that both latent KSHV infection and KapB expression enhanced autophagic flux via phosphorylation of the autophagy regulatory protein, Beclin. KapB was necessary for this effect, as infection with a recombinant virus that does not express the KapB protein did not induce Beclin phosphorylation or autophagic flux. Moreover, we showed that PB disassembly mediated by KSHV or KapB, depended on autophagy genes and the selective autophagy receptor NDP52/CALCOCO2 and that the PB scaffolding protein, Pat1b, co-immunoprecipitated with NDP52. These studies reveal a new role for autophagy and the selective autophagy receptor NDP52 in promoting PB turnover and the concomitant synthesis of inflammatory molecules during KSHV infection.

## Author summary

Kaposi's sarcoma-associated herpesvirus (KSHV) causes several forms of cancer including Kaposi's Sarcoma (KS). Multiple KSHV gene products contribute to inflammatory processes that sustain cancer cells. Processing bodies (PBs) are poorly characterized cellular

**Funding:** GKS was supported by a CIHR CGS-M scholarship and Scotia Scholars NSHRF graduate training award. ELC was supported by a Killam predoctoral scholarship, an NSERC CGS-M scholarship, and a Nova Scotia Graduate scholarship. MK was supported by a CSM graduate training award, a CIHR CGS-M scholarship, and a CIHR doctoral award. Operating funds to support this work derive from a CIHR Project Grant PJT-153210 to JAC. The funders had no role in study design, data collection and analysis, decision to publish, or preparation of the manuscript.

**Competing interests:** The authors have declared that no competing interests exist.

structures that suppress inflammation by interfering with the mRNAs that encode inflammatory mediator proteins. We previously showed that a viral protein known as KapB caused PBs to disappear in KSHV infected cells. Here, we reveal the mechanism of PB disappearance in infected cells, whereby KapB co-opts the cellular catabolic process of autophagy to direct PB disassembly.

This discovery centers autophagy in viral control of inflammation and provides yet another striking example of subversion of normal cellular processes by cancer-causing viruses.

## Introduction

Macroautophagy, hereafter referred to as autophagy, is an evolutionarily conserved degradative process that ensures the continuous recycling of material to maintain cellular homeostasis [1–3]. Autophagy begins with the formation of a phagophore that elongates and encloses a portion of the cytoplasm to form a double-membrane organelle, the autophagosome, which fuses with lysosomes to form autolysosomes, where contents are degraded [4]. Under basal conditions, autophagic activity ensures the clearance of damaged macromolecular complexes and organelles, acting as an intracellular quality control system to maintain homeostasis [2]. Various changes to the cellular microenvironment, including nutrient scarcity, hypoxia, and endoplasmic reticulum (ER) stress can upregulate autophagy, reprioritizing resources. This provides the cell with the building blocks needed to synthesize macromolecules during adverse and stressful conditions, thereby promoting cell survival [2,5].

Although previously considered a non-selective bulk degradative pathway of cytoplasmic recycling, recent work revealed cargo targeting to autophagosomes through a process known as selective autophagy [6,7]. Selective autophagy can target specific cellular components, including organelles (e.g mitochondria, peroxisomes, ER) and protein complexes (e.g.ribosomes, focal adhesion complexes, protein aggregates) for degradation, fine tuning cellular quality and quantity control in the process [5,6,8]. In selective autophagy, target cargo is either selected directly, by binding to lipidated microtubule-associated protein 1 light chain 3 (LC3) protein (LC3-II)/Atg8, or via a receptor that mediates cargo recognition. These molecular bridges are termed selective autophagy receptors (SARs), and their role is to link the selected cargo to lipidated LC3-II molecules anchored in the expanding phagophore [9]. Some of the best characterized SARs include p62/SQSTM1, valosin containing protein (VCP), optineurin (OPTN), neighbor of BRCA1 gene 1 (NBR1), nuclear dot protein 52 (NDP52/calcium binding and coiled-coil domain protein, CALCOCO2) and its paralogs Tax1-binding protein 1 (TAX1BP1/CALCOCO3) and TAX1BP3/CALCOCO1 [6,10–15]. Of these, OPTN, p62, NDP52 and paralogs, and NBR1 are members of the sequestosome-like family and contain several conserved motifs including a LC3-interacting region (LIR) and a ubiquitin-binding domain (UBD) to mediate cargo selection via binding ubiquitin chains present on the cargo surface [7,16–18].

While SARs play important homeostatic roles in maintaining and at times magnifying the turnover of organelles and aggregates via cargo selection, an emerging function of these receptors is to localize the initiation of autophagy and modulate cellular signalling [19]. SARs p62 and NDP52 both promote autophagosome formation at the cargo site through direct interactions with autophagic initiating complexes [20–22]. SARs also play important roles in fine-tuning innate immune responses by targeting key signaling platforms for degradation [23–28]. For example, NDP52 is required for the autophagic degradation of interferon regulatory factor 3 (IRF3) and mitochondrial antiviral signaling protein (MAVS), which supports the resolution

phase of type I interferon signaling [25,29,30]. Selective autophagy can also degrade invading pathogens in a process called xenophagy [31]. During bacterial xenophagy, NDP52 plays a dual role, targeting bacteria to nascent autophagosomes and promoting autophagosome maturation to destroy the pathogen [32]. However, the precise role for selective autophagy in the xenophagy of viruses (virophagy) is less clear, as both proviral and antiviral roles have been attributed to SARs [33–35] and viruses can enhance or limit SAR function accordingly to promote their replication [36].

Kaposi's sarcoma-associated herpesvirus (KSHV) is the infectious cause of two B-cell malignancies and the endothelial cell cancer, Kaposi's sarcoma (KS), which features aberrant angiogenesis, tortuous leaky blood vessels, and inflammation [37–39]. Infection with KSHV has two forms, a latent phase and a lytic phase and it is the latent phase that predominates in KS tumours [40–42]. Like many viruses, KSHV manipulates the autophagic machinery. During latency, the viral FLICE inhibitory protein (v-FLIP) binds Atg3 to suppress the LC3 lipidation process during autophagosome biogenesis [43]. Latently infected cells also express a viral cyclin D homolog (v-cyclin) that triggers DNA damage leading to autophagy and cellular senescence [44], facilitating the production and non-conventional secretion of pro-inflammatory cytokines/chemokines such as IL-1β, IL-6 and CXCL8 [45]. KSHV manipulation of autophagic flux prevents xenophagy, limits interferon responses, and enhances inflammatory cytokine production. Precisely how KSHV fine-tunes autophagy during latency to balance proviral and antiviral activities and promote inflammation during chronic viral infection is not clear.

Processing bodies (PBs) are ubiquitous, cytoplasmic, ribonucleoprotein (RNP) granules that regulate expression of many cytokine RNA transcripts, making PBs important sites of inflammatory control. PBs contain enzymes involved in mRNA turnover including those that mediate mRNA decapping (Dcp2; co-factors Dcp1a, Pat1b and Edc4/Hedls), 5'-3' exonucleolytic degradation (5'-3' exonuclease Xrn1 and RNA helicase Rck/DDX6) and some components of the RNA-induced silencing complex [36,46–52]. mRNAs are routed to PBs for decay and/or repression by RNA-binding proteins (RBPs) through recognition of common sequence elements [47]. AU-rich elements (AREs) are destabilizing RNA regulatory elements found in the 3' untranslated regions (UTRs) of ~8% of cellular transcripts and are responsible for targeting RNAs to PBs [53–55]. When bound by destabilizing RBPs, ARE-containing RNAs are directed to PBs and suppressed [56–58]. Our group and others showed that PB size and number correlate with the constitutive decay or translational suppression of ARE-mRNAs; when PBs are dispersed, this suppression is reversed [59–63]. Because ARE-mRNAs encode potent regulatory molecules such as inflammatory cytokines, PBs are important post-transcriptional sites of regulation that fine-tune the production of inflammatory cytokines whose transcripts contain AREs, including IL-6, CXCL8, IL-1β, and TNF [54,64–70].

PBs are dynamic structures that are continuously assembled and disassembled, yet relatively little is known about how their activity is regulated. This is despite observations that stimuli that activate the stress-responsive p38/MAPKAPK2 (MK2) kinase pathway, as well as many virus infections, elicit PB disassembly and a concomitant reduction in ARE-mRNA suppression to promote inflammatory molecule production [59,60,63,71–73]. We have previously shown that KSHV causes PB disassembly during latent infection, and that the viral protein Kaposin B (KapB) induces PB disassembly while enhancing the production of an ARE-containing reporter [60,71,74]. These data support the notion that PB disassembly is likely an important contributor to inflammation associated with KS [60,75]. Although we know that KapB binds and activates the kinase MK2, and that MK2 is an important component of the mechanism of KapB-mediated PB disassembly and ARE-mRNA stabilization, we do not precisely understand how PB loss is elicited by KapB [60,76]. The observation that MK2 can

phosphorylate the autophagy regulatory protein Beclin 1 (hereafter referred to as Beclin) to increase autophagic flux in response to nutrient scarcity [77] suggested to us that KapB may drive autophagic flux through MK2. We now show that ectopic expression of the KSHV KapB protein enhances autophagic flux via the phosphorylation of Beclin. We show that KSHV infection enhances autophagic flux and Beclin phosphorylation in a KapB-dependent manner, and that during KSHV latency, MK2 is required for autophagic flux increases. Together these data reveal an additional layer of complexity in viral regulation of autophagy during KSHV latency. We show that both KSHV infection and KapB require autophagy proteins, including the selective autophagy receptor NDP52, to induce PB disassembly and that KapB requires NDP52 to elevate selected ARE-containing mRNA transcripts. PB turnover is likewise elicited by potent chemical induction of autophagy and this process is also dependent on autophagy genes and NDP52, further underscoring the importance of autophagic flux in regulating the basal turnover of PBs. Finally, we show that NDP52 immunoprecipitates with the PB scaffold-ing protein, Pat1b, providing a molecular basis for the recognition of a PB component by NDP52. These data forge a new connection between the selective autophagy receptor NDP52 and cytoplasmic PBs and show that selective autophagy of PBs is promoted by the viral protein KapB during KSHV infection.

## Results

### KapB induces autophagic flux

KapB expression activates the MK2 kinase pathway [60,76]. Since MK2 phosphorylates the essential autophagy inducer Beclin in response to amino acid starvation [77–79], we ques-tioned whether KapB expression could promote autophagic flux. Essential for autophagosome expansion is the lipidation of LC3 to form LC3-II, permitting its incorporation into autopha-gosome membranes where it is degraded along with autophagosomal contents in the final step of autophagy [80]. Since p62 is a selective autophagy receptor (SAR) that delivers target cargo to the autophagosome, p62 is also degraded during the final step of autophagy. If autophagic flux is elevated, blocking autophagic flux with an inhibitor that interferes with autophago-some-lysosome fusion (e.g. Bafilomycin A1) should result in a greater accumulation of both p62 and LC3-II than at baseline [80,81]. To measure autophagic flux during KapB expression, we examined levels of LC3-II and p62 with and without treatment with Bafilomycin A1 (BafA1) compared to controls. Human umbilical vein endothelial cells (HUVECs) were trans-duced with lentiviral vectors that express KapB or an empty vector control and treated for increasing times with BafA1. As expected, LC3-II and p62 accumulated in the presence of BafA1 treatment, and both proteins showed greater accumulation after BafA1 treatment in KapB-expressing cells than equivalently treated vector controls (Fig 1A). When KapB-express-ing cells were treated for 4 h with BafA1, the amount of LC3-II and p62 increased two- and four-fold, respectively (Fig 1A). These data indicate that autophagic flux was enhanced in KapB-expressing cells compared to vector controls. [80] We also measured the formation of LC3 puncta as an indicator of autophagic flux by staining KapB-expressing cells treated with BafA1 for endogenous LC3 [82,83]. Untreated KapB-expressing HUVECs displayed similar LC3 puncta area to control cells; however, LC3 puncta area significantly increased after BafA1 treatment in KapB-expressing cells and not in control cells, further suggesting that KapB expression enhanced autophagic flux (Fig 1B).

### KapB-mediated PB disassembly requires Atg5 and Atg14

Our group previously described that MK2 activation was an important component of KapB-mediated PB loss [60]; however, these experiments did not reveal the precise mechanism for

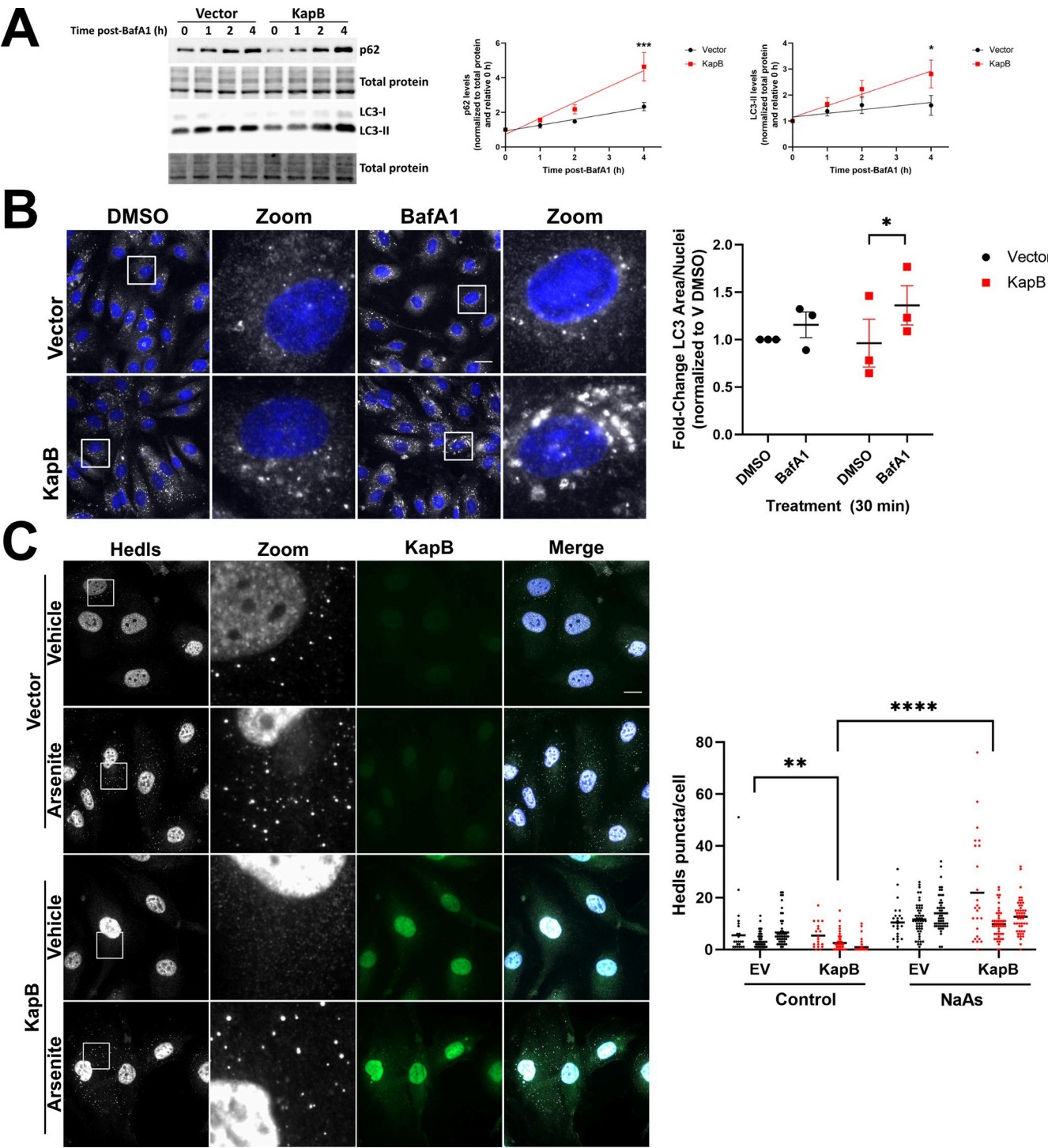

**Fig 1. Kaposin B increases autophagic flux.** Primary human umbilical vein endothelial cells (HUVECs) were transduced with recombinant lentiviruses expressing either Kaposin B (KapB) or an empty vector control and selected with blasticidin (5 μg/mL). A: Cells were treated with Bafilomycin A1 (BafA1, 10 nM) or a vehicle control (DMSO) for the indicated times prior to harvest in 2X Laemmli buffer. Protein lysates were resolved by SDS-PAGE and immunoblot was performed for p62 and LC3. Samples were quantified using Image Lab (BioRad) software and then normalized, first to total protein and then to their respective starting time points (0 h). Results were plotted in GraphPad and a linear regression statistical test was performed, ±SEM; n = 3, * = P<0.05, *** = P<0.001. B: Cells were treated with BafA1 for 30 min prior to fixation in methanol. Immunofluorescence was performed for LC3 (white) and DAPI (nuclei, blue). Scale bar = 20 μm. Total LC3 area per field was quantified by identifying LC3-positive puncta using CellProfiler and normalizing to the number of nuclei and the vector DMSO control. Results were plotted in GraphPad, a 2-way ANOVA was performed with a Šidák's multiple comparison test, matched data from each replicate was paired for comparison, ±SEM; n = 3, * = P<0.05. C: HUVECs were transduced with recombinant lentiviruses expressing either

KapB or an empty vector control and selected with blasticidin (5 μg/mL). Cells were treated with sodium arsenite (0.25 mM) or a vehicle control for 30 min prior to fixation in 4% paraformaldehyde and permeabilization in 0.1% Triton X-100. Scale bar = 20 μm. Hedls puncta were quantified using CellProfiler and presented as number of Hedls puncta per cell, all cells counted are displayed. Results were plotted in GraphPad and a 2-way ANOVA was performed on the main column effects with a Tukey's multiple comparison test, bar represents the mean; n = 3, ** = P<0.01, **** = P<0.0001.

this loss. Since PBs are dynamic RNP granules, we wanted to determine whether KapB-mediated PB loss was a result of enhanced disassembly of PBs or prevention of their *de novo* assembly. To do so, we utilized HeLa cells that express a Dox-inducible GFP-tagged version of the PB-resident protein, Dcp1a [49]. When fixed, GFP-positive puncta co-stain with endogenous PB proteins such as the RNA helicase DDX6/Rck and the decapping co-factor Hedls/Edc4, indicating that they represent PBs (S1 Fig). KapB was transfected into these cells either before or after inducing GFP-Dcp1a granule formation. When KapB was expressed prior to granule induction, Dcp1a puncta formed, although appearance of GFP-positive puncta was delayed compared to vector control cells (S1A Fig). In the presence of KapB, *de novo* PB assembly was not blocked; however, when KapB was expressed after PB induction, GFP-positive puncta were lost (S1B Fig), indicating that KapB expression induced PB disassembly. As further evidence that KapB does not prevent PB formation, we treated KapB-expressing HUVECs with sodium arsenite, a known inducer of PB assembly [84]. Sodium arsenite induced endogenous PB formation in KapB-expressing HUVECs to a similar extent to equivalently treated control cells (Fig 1C). Taken together, these results show that KapB expression caused PB disappearance by inducing PB disassembly, but KapB did not block PB assembly. We speculated that KapB may be utilizing a normal cellular pathway that mediates the turnover of RNPs or their components, such as autophagy.

To determine if autophagy was required for KapB-mediated PB disassembly, we first inhibited autophagic flux by independently silencing two genes in the macroautophagy pathway, Atg5 and Atg14 (S2A and S2B Fig). When either Atg5 or Atg14 were silenced in KapB-expressing HUVECs, PBs were restored to a level comparable to control cells (Fig 2A and 2B). We then examined the ability of KapB to mediate PB disassembly in Atg5 -/- mouse embryonic fibroblasts (MEFs), which [85] lack conversion of LC3-I to LC3-II (S2C Fig). KapB failed to disassemble PBs in Atg5 -/- MEFs but did disassemble PBs in matched wild-type controls (S2D Fig). Moreover, KapB expression resulted in a significant increase in Atg5 protein levels in HUVECs (S2A Fig), an observation that is consistent with other data that show that KapB promotes autophagy. Additionally, blocking autophagic degradation with BafA1 also restored PBs in KapB-expressing cells (Fig 2C). Together, these data indicate that KapB mediates PB disassembly by enhancing autophagy.

Basal autophagy can be upregulated by many different cell stimuli including starvation or inhibition of mTOR using Torin-1, hereafter referred to as Torin [86]. It was previously reported that rapamycin, an inhibitor of mTORC1, caused a loss of PBs in mammary epithelial cells [87]. We reasoned that if we inhibited both mTOR complexes mTORC1 and mTORC2 using Torin [88] to induce autophagy, PBs would disassemble. We observed that the number of Hedls/Edc4 puncta were significantly reduced after Torin treatment, indicating that chemical induction of autophagy also resulted in the disappearance of PBs (Fig 3A). To ensure that Torin treatment was inducing disassembly of PBs and not preventing assembly, HUVECs were treated with sodium arsenite to induce PB formation; like KapB, Torin treatment did not prevent the assembly of PBs under these conditions (Fig 3B).

Given these similarities, we wondered if KapB induced autophagy via inhibition of mTOR complexes mTORC1 and mTORC2. To test this, we performed immunoblotting for the phosphorylated form of the ribosomal protein S6 (Serine 235/236) [89] and the phosphorylated

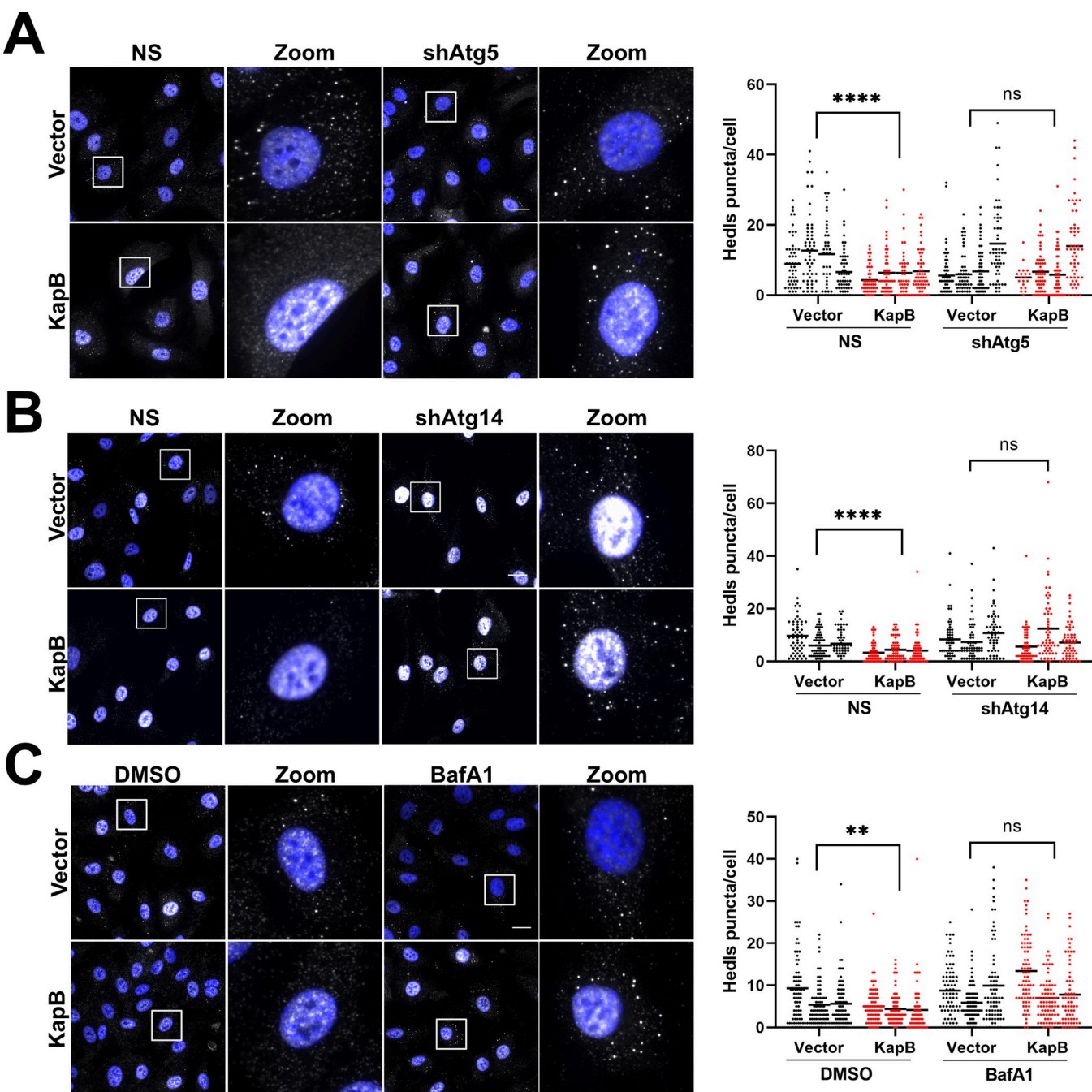

**Fig 2. Kaposin B requires Atg5 and Atg14 autophagy to disassemble PBs.** A & B: HUVECs were sequentially transduced: first with recombinant lentiviruses expressing either shRNAs targeting Atg5 (A) or Atg14 (B) (shAtg5, shAtg14) or a non-targeting control (NS) and selected with puromycin (1 μg/mL), and second with either KapB or an empty vector control and selected with blasticidin (5 μg/mL). Coverslips were fixed in 4% paraformaldehyde, permeabilized in 0.1% Triton X-100, and immunostained for the PB-resident protein Hedls (white) and nuclei (DAPI). Scale bar = 20 μm. Hedls puncta were quantified using CellProfiler and presented as number of Hedls puncta per cell, all cells counted are displayed. Results were plotted in GraphPad and a 2-way ANOVA was performed on the main column effects with a Tukey's multiple comparison test, bar represents the mean; n = 4 (A), n = 3 (B), **** = P<0.0001. C: HUVECs were transduced with recombinant lentiviruses expressing either KapB or an empty vector control and selected with blasticidin (5 μg/mL). Cells were treated with BafA1 for 30 min, fixed in 4% paraformaldehyde, permeabilized in 0.1% Triton X-100, and immunostained for the PB-resident protein Hedls (white) and nuclei (DAPI). Scale bar = 20 μm. Hedls puncta were quantified using CellProfiler and presented as number of Hedls puncta per cell, all cells counted are displayed. Results were plotted in GraphPad and a 2-way ANOVA was performed on the main column effects with a Tukey's multiple comparison test, bar represents the mean; n = 3, ** = P<0.01.

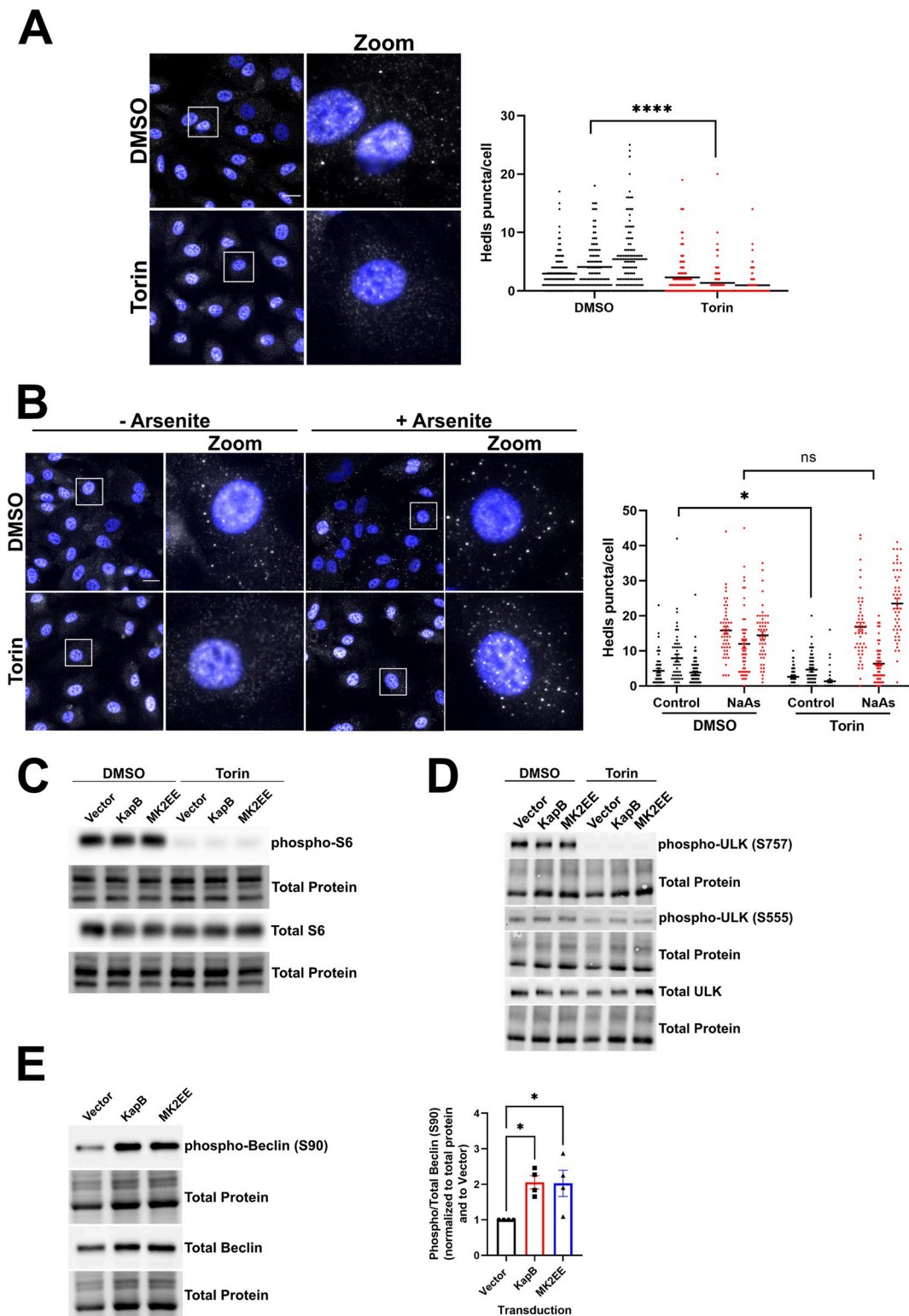

**Fig 3. Torin treatment disassembles PBs.** A: HUVECs were treated with either Torin (250 nM) or a DMSO control for 2 h prior to fixation in 4% paraformaldehyde. Samples were permeabilized in 0.1% Triton X-100. Immunofluorescence was performed for Hedls (PBs, white) and DAPI (nuclei, blue). Scale bar = 20 μm. Hedls puncta were quantified using CellProfiler and presented as number of Hedls puncta per cell, all cells counted are displayed. Results were plotted in GraphPad and a 2-way ANOVA was performed on the main column effects with a Tukey's multiple comparison test, bar represents the mean;

n = 3, **** = P<0.0001. B: HUVECs were treated with either Torin (250 nM) or a DMSO control for 90 min prior to the addition of sodium arsenite (0.25 mM) for 30 min. Cells were fixed in 4% paraformaldehyde and permeabilized in 0.1% Triton X-100. Immunofluorescence was performed for Hedls (PBs, white) and DAPI (nuclei, blue). Scale bar = 20 μm. Hedls puncta were quantified using CellProfiler and presented as number of Hedls puncta per cell, all cells counted are displayed. Results were plotted in GraphPad and a 2-way ANOVA was performed on the main column effects with a Tukey's multiple comparison test, bar represents the mean; n = 3, * = P<0.05. C: HUVECs were transduced with recombinant lentiviruses expressing either KapB, constitutively active MK2 (MK2EE), or an empty vector control and selected with blasticidin (5 μg/mL). Cells were treated with either Torin (250 nM) or a DMSO control for 2 h prior to harvest in 2X Laemmli buffer. Protein lysates were resolved by SDS-PAGE and immunoblot was performed for phospho-S6 (Ser235/236) and total S6. D: HUVECs were transduced and treated as in C and harvested in 2X Laemmli buffer. Protein lysates were resolved by SDS-PAGE and immunoblot was performed for phospho-ULK (Ser757), phospho-ULK (Ser555), and total ULK. E: HUVECs were transduced as in C and harvested in 2X Laemmli buffer. Protein lysates were resolved by SDS-PAGE and immunoblot was performed for phospho-Beclin (Ser90) and total Beclin. Samples were quantified by normalizing total and phospho-Beclin protein levels to the total protein in each lane and then using Image Lab (BioRad), normalizing to the vector control, and then dividing phospho- over total Beclin. Results were plotted in GraphPad and a one-way ANOVA was performed with a Dunnett's test, ±SEM; n = 4, * = P<0.05.

forms of the Unc-51 like autophagy activating kinase (ULK1; Serine 757/555) component of the ULK autophagy initiation complex. Dephosphorylation of both proteins occurs when mTOR activity is inhibited by Torin and autophagy is initiated [90,91]. Unlike Torin, KapB expression did not decrease phosphorylation of S6 (Fig 3C) or ULK (Fig 3D). We also expressed a constitutively active form of MK2 in which two threonine residues that are normally phosphorylated by the upstream kinase p38, have been substituted with glutamic acid residues (T205E/T317E, called MK2EE) [92–94] and determined the effect on phospho-S6 and phospho-ULK. We previously demonstrated that MK2EE caused PB disassembly [60] and others showed it increased autophagic flux after starvation [77]. Like KapB, MK2EE expression did not decrease phosphorylation of S6 (Fig 3C) or ULK (Fig 3D). These data indicate KapB does not inhibit mTOR activity to induce autophagy. Since KapB can bind and activate MK2, we next wondered if KapB expression would mediate phosphorylation of Beclin at Serine 90, as shown by others for MK2EE [68]. Like MK2EE, KapB induced phosphorylation of Beclin at Serine 90 (Fig 3E), indicating that induction of autophagy via KapB expression is mediated by activation of Beclin. Taken together, these data show that although both KapB expression and Torin treatment caused PB disassembly in an autophagy-dependent manner, the mechanism used to upregulate autophagic flux differed in each case.

## Autophagic machinery is required for KapB-mediated increases in ARE-containing cytokine transcripts

PBs are nodes of post-transcriptional mRNA regulation and microscopically visible PBs are associated with ARE-mRNA decay or suppression [51,59,60,62,63,95,96]. However, under basal conditions there is limited transcription of most ARE-containing cytokine or chemokine transcripts, making their measurement difficult. Rather than using an inflammatory stimulus such as lipopolysaccharide to activate transcription of ARE-containing cytokines, we chose to activate their transcription in a biologically relevant manner by mimicking the inflammatory environment of the KS lesion. KapB-expressing or control HUVECs were treated with 0.22 μm-filtered conditioned media from KSHV latently infected cells, expected to contain inflammatory cytokines e.g. TNF that would then further activate cytokine gene transcription [97]. We determined the RNA level of endogenous ARE-containing transcripts in KapB-expressing cells before and after the conditioned media treatment and compared these to equivalently treated control HUVECs (Fig 4A). In the context of KapB expression, IL-1β mRNA was significantly elevated compared to equivalently treated controls before and after conditioned media treatment (Fig 4A). These data suggest that this ARE-containing transcript

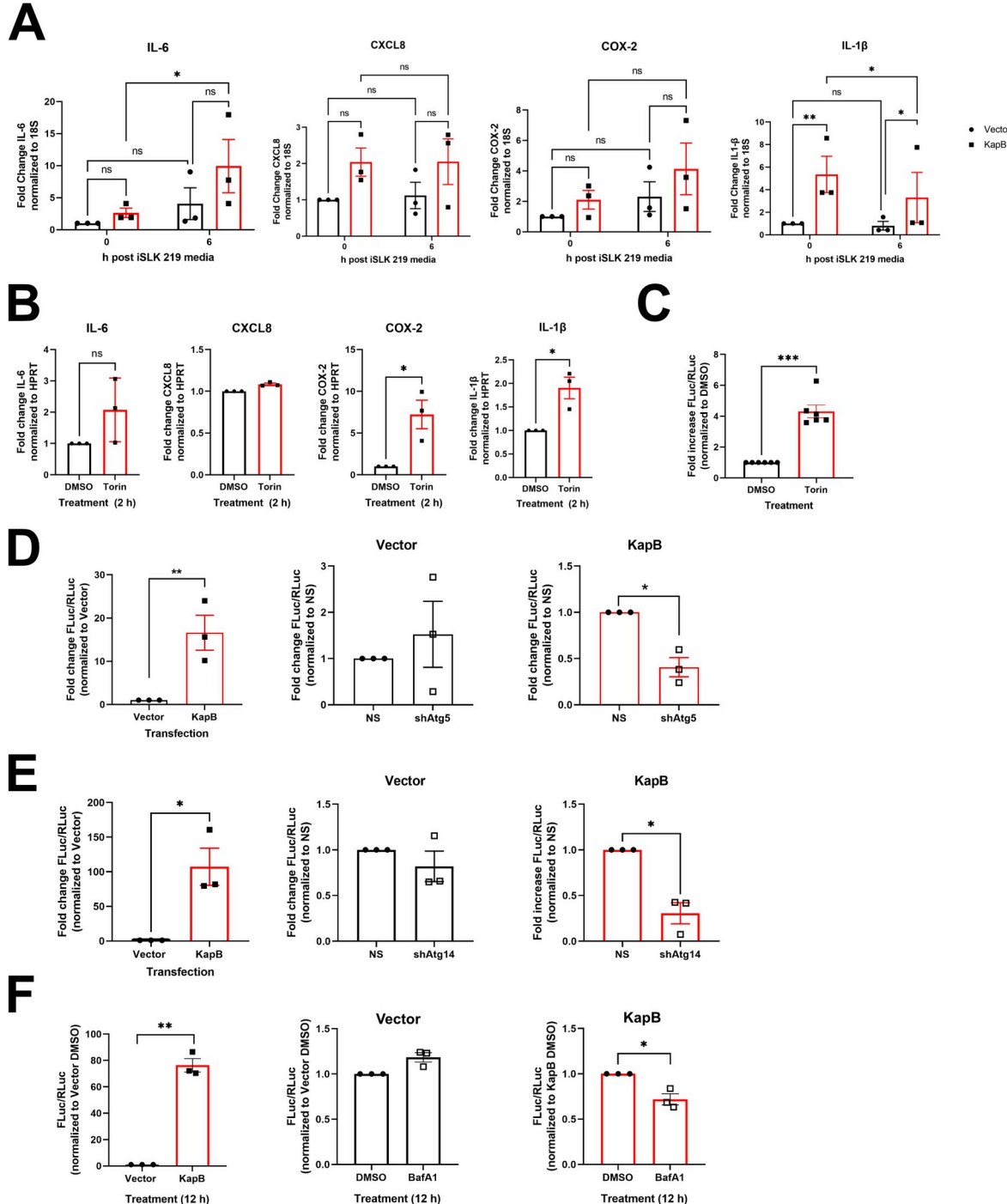

**Fig 4. KapB-mediated ARE-mRNA reporter expression requires autophagy.** A: HUVECs were transduced with recombinant lentiviruses expressing either KapB or an empty vector control and selected with blasticidin (5 µg/mL). Conditioned media harvested from rKSHV.219 latently infected iSLK cells was used to mimic the KS lesion microenvironment and induce the transcription of cytokines for 6 h prior to lysis for total RNA (normal media was used for the 0 h time point). Transcript levels were quantified by qPCR and normalized to 18S as a reference gene. Data is represented as the fold change in target transcript expression relative to the untreated vector control and was quantified using the ΔΔCq method. Results were plotted in GraphPad, a 2-way ANOVA with a Tukey's multiple comparisons test was performed, ±SEM; n = 3, * = P<0.05, ** = P<0.01. B: HUVECs were treated with Torin (250 nM) or a DMSO control for 2 h prior to lysis for total RNA. Transcript levels were quantified by qPCR and normalized to HPRT as a reference gene. Data is represented as the fold change in target transcript expression relative to the untreated vector control and was quantified using the ΔΔCq method. An unpaired t-test was performed, ±SEM; n = 3, * = P<0.05. C: HeLa Tet-Off cells were co-transfected with expression plasmids for an ARE-containing

firefly luciferase plasmid (pTRE-FLuc-ARE) and a stable renilla luciferase plasmid (pTRE-RLuc). 36 h post transfection, doxycycline (Dox) was added to halt reporter gene transcription of both luciferase reporters, at the same time Torin (250 nM) or DMSO were added; 12 h after Dox addition, lysates were harvested in passive lysis buffer (Promega). Luciferase activity for both FLuc and RLuc was analyzed using the Dual-Luciferase Reporter Assay (Promega) and normalized (FLuc/RLuc) relative luciferase was calculated in relative light units (RLUs). Results were plotted using GraphPad, an unpaired t-test was performed, ±SEM; n = 6, *** = P<0.001. D&E: HeLa Tet-Off cells were transduced with recombinant lentiviruses expressing either shRNAs targeting Atg5 or Atg14 (shAtg5, shAtg14) or a non-targeting control (NS) and selected with puromycin (1 μg/mL). After selection, cells were co-transfected and Dox treatment was performed as described in C, except that co-transfection also included an expression plasmid for KapB or an empty vector control. Results were plotted using GraphPad, an unpaired t-test was performed, ±SEM; n = 3, * = P<0.05, ** = P<0.01. F: Cells were co-transfected as in E and BafA1 (10 nM) was added at the same time as Dox. Results were plotted in GraphPad and a Student's t-test was performed, ±SEM; n = 3, * = P<0.05, ** = P<0.01.

is subject to constitutive decay in PBs that is alleviated by KapB. Moreover, it suggests that KapB-mediated PB disassembly enhances the baseline RNA levels as well as in the context of inflammatory stimuli typical of the KS microenvironment. IL-6 transcript levels significantly increased in KapB-expressing cells after but not before conditioned media treatment (Fig 4A). These data suggest that the IL-6 transcript requires an inflammatory transcriptional stimulus (e.g. KS microenvironment) combined with KapB-mediated PB disassembly for a significant increase. RNA transcripts encoding COX-2 and the chemokine CXCL8 were not significantly elevated by conditioned media treatment alone, nor were they elevated by KapB expression alone, or by the combination of media treatment and KapB (Fig 4A). These data suggest that additional factors contribute to elevated RNA levels of these ARE-containing RNA transcripts. The observation that different cytokine RNA transcripts respond differently to alterations in PB dynamics was also made by others [98]. Together, these data suggest that some cytokine transcripts respond to KapB-mediated PB disassembly with elevated levels either at baseline or upon transcriptional induction, an effect that likely results from reduced transcript decay in PBs (Fig 4A). In addition to KapB, we also treated HUVECs with Torin. This treatment enhanced steady-state levels of IL-1β and COX-2 (Fig 4B), suggesting that the induction of autophagy by an alternative mechanism that also culminated in PB disassembly was sufficient to elevate certain ARE-mRNA transcripts. This suggests that cytokine mRNAs that shuttle to PBs are relieved from constitutive turnover by stimuli that upregulate autophagic flux and cause PB disassembly.

To support these findings and to simplify further studies we utilized a luciferase reporter assay that we developed, described in [74]. Using this assay, we previously showed that KapB-mediated PB loss correlated with enhanced luminescence of an ARE-containing firefly luciferase (FLuc) reporter because its rapid turnover was reversed and FLuc translation was enhanced [60,74]. Briefly, HeLa cells were co-transfected with a FLuc construct containing the destabilizing AU-rich element from the *CSF2* gene in its 3'UTR and a Renilla luciferase (RLuc) construct with no ARE; transcription of both reporter constructs was then stopped by the addition of doxycycline. In control (vector transfected) cells, the FLuc-ARE mRNA decayed rapidly and FLuc luminescence relative to the RLuc transfection control was low. Torin treatment caused a significant increase in FLuc/RLuc relative luminescence compared to DMSO-treated control cells, supporting our RT-qPCR findings (Fig 4C) and suggesting that enhanced autophagy reverses the constitutive turnover/suppression of the ARE-mRNA reporter. Furthermore, KapB expression also caused a significant increase in FLuc/RLuc relative luminescence compared to empty vector control cells (Fig 4D) as previously shown in [60]. When autophagic flux was perturbed by either Atg5 or Atg14 silencing (S3A and S3B Fig), KapB-expressing cells displayed significantly decreased luminescence compared to KapB-expressing controls (Fig 4D and 4E). Likewise, when KapB-expressing cells were treated with BafA1, relative luminescence was significantly decreased compared to KapB-expressing untreated control cells

(Fig 4F). Together, these data showed that autophagic pathways are required for KapB-mediated increases in cytokine mRNA levels and the enhanced expression of an ARE-mRNA reporter.

## Dcp1a protein levels are decreased by KapB and Torin treatment

Both KapB and Torin mediated PB disassembly but did not prevent *de novo* PB formation. We reasoned that KapB and Torin could have this effect because they cause the autophagic degradation of the entire PB granule. Alternatively, since some PB proteins have important scaffolding roles that hold granules together (EDC4/Hedls, DDX6/Rck, Pat1b, Dcp1a) [99], PB disassembly could be induced by KapB or Torin if one or more key PB proteins was missing due to autophagic degradation. To differentiate between these two scenarios, we immunoblotted for PB proteins after Torin treatment and in the context of KapB expression (Fig 5). Immunoblotting revealed that steady-state levels of Dcp1a were reduced in Torin-treated (Fig 5A) and KapB-expressing cells (Fig 5B); however, steady-state levels of the other PB-resident proteins we tested were not affected (Fig 5A and 5C). Moreover, KapB-mediated Dcp1a decreases were reversed by BafA1 treatment (Fig 5B) and steady-state levels of the SAR, p62, included as a measure of autophagic flux, were likewise restored (Fig 5B). Together, these data suggest that autophagy induction with Torin or KapB expression promote the loss of the PB-resident protein Dcp1a, but not all PB proteins. Silencing *Atg5* prevented Dcp1a loss in both KapB and control cells (Fig 5D), suggesting that autophagy is required for the turnover of Dcp1a and that KapB accelerates this autophagic turnover.

## KapB-mediated PB disassembly and ARE-mRNA stabilization require the selective autophagy receptor NDP52

After establishing a link between KapB expression, autophagy, and PB disassembly, we wondered if SARs were involved in PB disassembly. One previous study observed that the NDP52 SAR partially co-localized with some PB proteins [100]. For this reason, we used RNA silencing to investigate whether NDP52, p62, VCP, NBR1, or OPTN were required for KapB-mediated PB disassembly. Despite trying several different shRNA sequences, our attempts to silence NBR1 were unsuccessful and shRNAs targeting VCP induced significant HUVEC cell death (S4A and S4B Fig); therefore, the role of these molecules in KapB-mediated PB disassembly could not be determined. Knockdown of NDP52, p62, and OPTN was validated in control and KapB-expressing cells (S4C–S4E Fig). Silencing of p62 or OPTN did not restore PBs in KapB-expressing cells whereas NDP52 knockdown significantly restored PBs (Figs 6A and S4F). These data pinpointed an important role for NDP52 and selective autophagy in KapB-mediated PB disassembly. We then tested whether NDP52 silencing could prevent Torin-mediated PB disassembly and found that PBs could not be disassembled in cells lacking NDP52 (Figs 6B and S4G). Since steady-state levels of the decapping cofactor, Dcp1a, decreased in response to KapB infection and Torin treatment (Fig 5), we examined the steady-state level of Dcp1a after Torin with and without NDP52 silencing (Fig 6C). Although Dcp1a was significantly diminished after Torin treatment, NDP52 silencing did not restore Dcp1a to control levels. We decided to consider another PB protein, Pat1b, a decapping cofactor known to interact with several other PB proteins including Dcp1a, DDX6, and Hedls [52]. Immunoblotting revealed that Torin treatment significantly diminished steady-state levels of Pat1b, and that this decrease was restored to control levels when Torin treatment was performed in NDP52-silenced cells (Fig 6C). These data suggest that levels of the PB component proteins Pat1b, and to a lesser extent, Dcp1a, are regulated by NDP52-mediated autophagic degradation.

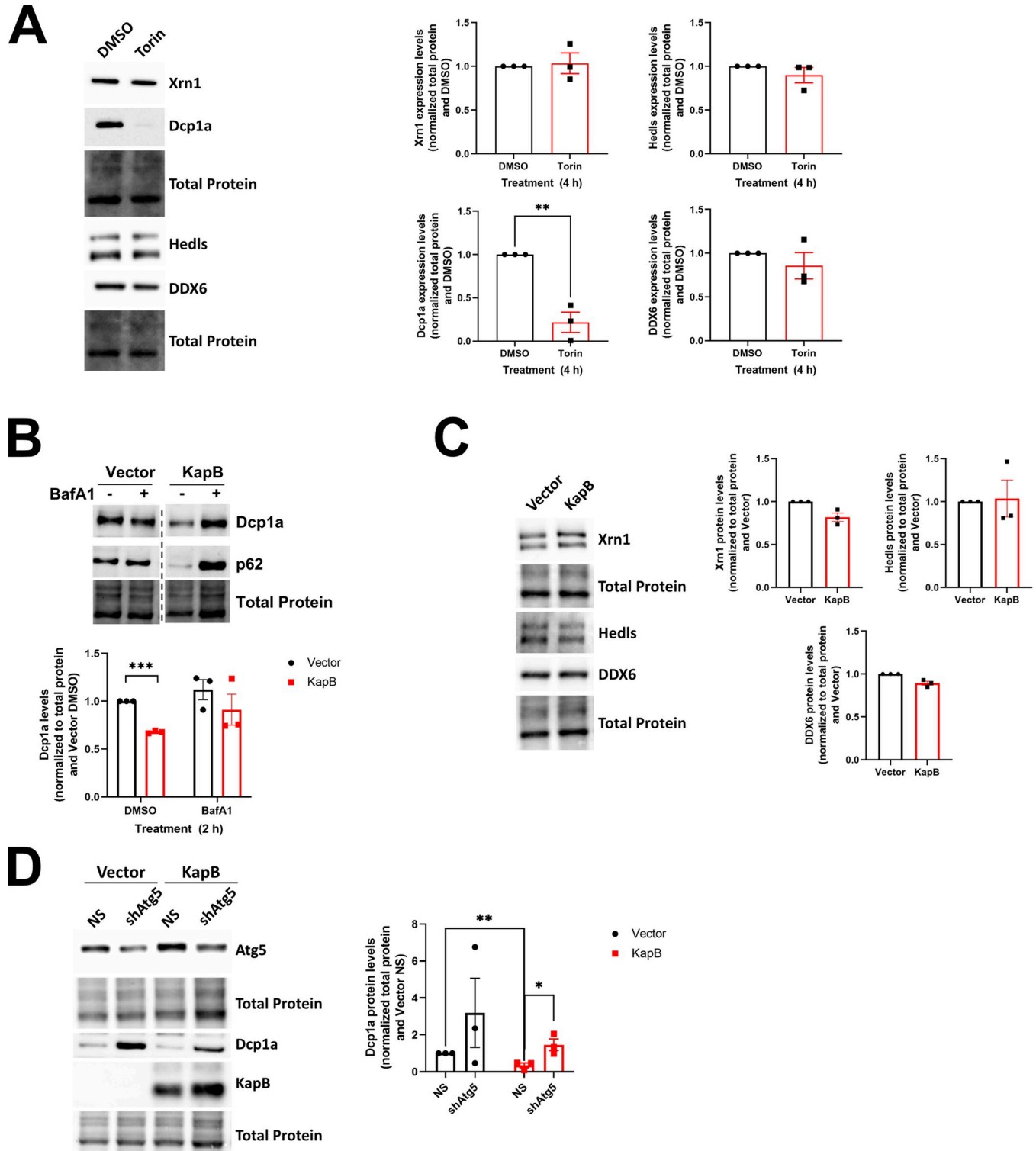

**Fig 5. Dcp1a protein levels are decreased by KapB expression and Torin treatment.** A: HUVECs were treated with DMSO or Torin (250 nM) for 4 h prior to harvest. Samples were lysed in 2X Laemmli buffer and resolved by SDS-PAGE before immunoblotting for Xrn1, EDC4/Hedls, Dcp1a, and DDX6. Samples were quantified by normalizing the PB resident protein levels to the total protein in each lane and then the DMSO control using ImageLab (BioRad). Results were plotted in GraphPad and a one-way ANOVA was performed ±SEM; n = 3, ** = P<0.01. B: HUVECs were transduced with recombinant lentiviruses expressing either KapB or an empty vector control and selected with blasticidin (5 μg/mL). Cells were treated with DMSO or Bafilomycin A1 (BafA1, 10 nM) for 4 h prior to harvest in 2X Laemmli buffer. Samples were resolved by SDS-PAGE and immunoblot was performed for Dcp1a or p62 (autophagy marker).

Samples were quantified by normalizing Dcp1a protein levels to the total protein in each lane using Image Lab (BioRad) and then to the vector DMSO control. Representative blot is from the same membrane, hashed line indicates skipped lanes. Results were plotted in GraphPad and a 2-way ANOVA was performed, ±SEM; n = 3, *** = P<0.001. C: HUVECs were transduced with recombinant lentiviruses expressing either KapB or an empty vector control and selected with blasticidin (5 µg/mL). Samples were harvested in 2X Laemmli buffer, resolved by SDS-PAGE and immunoblot was performed for Xrn1, Hedls/EDC4, or DDX6. Samples were quantified by normalizing PB protein levels to the total protein in each lane using Image Lab (BioRad) and then to the vector control. Results were plotted in GraphPad. D: HUVECs were sequentially transduced: first with recombinant lentiviruses expressing either shRNAs targeting Atg5 (shAtg5) or a non-targeting control (NS) and selected with puromycin (1 µg/mL), and second with either KapB or an empty vector control and selected with blasticidin (5 µg/mL). Samples were harvested in 2X Laemmli buffer and resolved by SDS-PAGE. Immunoblot was performed for Dcp1a, Atg5, and KapB. Samples were quantified by normalizing Dcp1a protein levels to the total protein in each lane using Image Lab (BioRad) and then to the vector NS control. Results were plotted in GraphPad and a 2-way ANOVA was performed, ±SEM; n = 3, * = P<0.05, **P = <0.01.

We also examined whether these SARs were required for KapB-enhanced relative luminescence in our ARE-mRNA reporter assay. We validated silencing of NDP52, p62, and OPTN in HeLa cells (S5A–S5C Fig). We found that p62 or OPTN knockdown had no effect on KapB-mediated ARE-containing reporter luminescence, but NDP52 silencing decreased the signal compared to non-targeting, KapB-expressing controls (Fig 6D), supporting a role for NDP52 in autophagy-mediated ARE-mRNA reporter increases. We then performed an NDP52 complementation experiment. We first silenced endogenous NDP52 in control or KapB-expressing HUVECs using shRNAs targeting the 3'UTR of NDP52 and verified its knockdown (S6A Fig). Next, NDP52 expression was restored by overexpressing a fluorescently tagged NDP52 construct. Complementation of NDP52-silenced HUVECs with NDP52 restored KapB-mediated PB disassembly compared to vector controls (Figs 6E–6F, S6B and S6C). Notably, complementation with NDP52 in control cells that lacked KapB failed to induce PB disassembly, suggesting that a KapB- or Torin-mediated autophagy increase is an important component of the PB disassembly phenotype. Taken together, these data indicate that NDP52 is required for KapB- and Torin-mediated PB disassembly and KapB-mediated ARE-mRNA stabilization.

## KapB is required for enhanced autophagic flux during latent KSHV infection

Autophagy regulation during KSHV latency is complex as the viral gene products v-FLIP and v-cyclin also modulate autophagic flux in addition to KapB [43,44]. To simplify our study of autophagy and PBs, we initially used a reductionist approach, expressing only KapB in HUVECs. However, to confirm the relevance of our study to KSHV infection, we performed a study in KSHV-infected HUVECs. We infected HUVECs with rKSHV.219 [101] for 96 hours to permit latency establishment and then treated the cells with either BafA1 or a vehicle control for 30 minutes prior to staining infected cells for LC3 puncta as a measure of autophagic flux. We observed a dramatic increase in LC3 puncta in KSHV infected cells after BafA1 treatment when compared to mock infection controls (Fig 7A). These data demonstrate that autophagic flux was upregulated during KSHV latency despite the concurrent expression of vFLIP, an inhibitor of autophagy, during this infection phase [43]. In parallel, we immunoblotted infected cell lysates and observed that KSHV increased phosphorylation of Ser90 of Beclin (Fig 7B), consistent with earlier observations of Ser90 phosphorylation after ectopic expression of KapB (Fig 3D). To confirm if KapB was necessary for upregulated autophagic flux after KSHV infection, we used a recombinant KSHV that does not express KapB (delB BAC16) and matched wild-type control (BAC16 KSHV), described in [102]. We infected HUVECs with WT BAC16 KSHV or delB BAC16 virus for 96 hours to permit latency establishment. WT BAC16 KSHV latency also induced LC3 puncta formation and phosphorylation of Ser90 of Beclin, consistent with our result using the KSHV.219 virus, whereas infection with delB BAC16 virus failed to enhance autophagic flux or phosphorylation of Beclin at Ser90 (Fig 7C and 7D). We confirmed these data in iSLKs, a KSHV producer cell line known for tight control

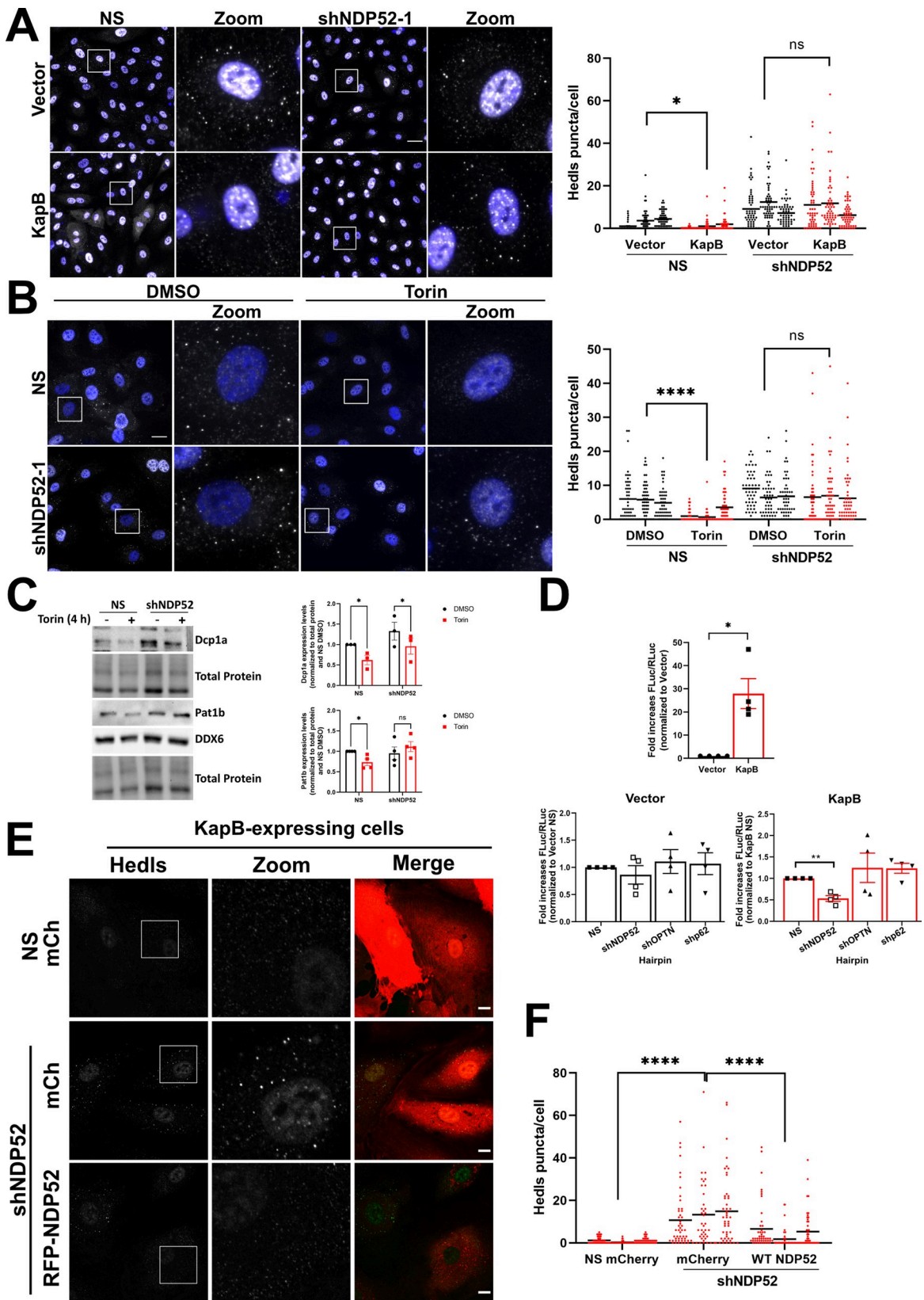

**Fig 6. KapB-mediated PB disassembly and ARE-mRNA reporter expression require the selective autophagy receptor NDP52.** A: HUVECs were sequentially transduced, first with recombinant lentivirus expressing KapB or an empty vector control and selected with blasticidin (5 µg/mL), and second with recombinant lentivirus expressing shRNAs targeting NDP52 or a non-targeting control (NS) and selected with puromycin (1 µg/mL). Coverslips were fixed in 4% paraformaldehyde, permeabilized in 0.1% Triton X-100 and immunostained for Hedls (PBs; white), DAPI (nuclei, blue). Scale bar = 20 µm. Hedls puncta were quantified using CellProfiler and presented as number of Hedls puncta per cell, all cells counted are displayed. Results were plotted in GraphPad and a 2-way ANOVA was performed on the main column effects with a Tukey's multiple comparison test, bar represents the mean; n = 3, * = P<0.05. B: HUVECs were transduced with recombinant lentivirus expressing an shRNA targeting NDP52 and selected with puromycin (1 µg/mL). Cells were treated with Torin (250 nM) or a DMSO control for 2 h prior to fixation in 4% paraformaldehyde and permeabilization in 0.1% Triton X-100. Samples were immunostained for Hedls (PBs; white), DAPI (nuclei, blue). Scale bar = 20 µm. Hedls puncta were quantified using CellProfiler and presented as number of Hedls puncta per cell, all cells counted are displayed. Results were plotted in GraphPad and a 2-way ANOVA was performed on the main column effects with a Tukey's multiple comparison test, bar represents the mean; n = 3, **** = P<0.0001. C: HUVECs were transduced with recombinant lentiviruses expressing either an shRNA targeting NDP52 or a non-targeting control (NS) and selected with puromycin (1 µg/mL). Cells were treated with DMSO or Torin (250 nM) for 4 h prior to harvest in 2X Laemmli buffer. Samples were resolved by SDS-PAGE and immunoblot was performed for Dcp1a, Pat1b or DDX6. Samples were quantified by normalizing Dcp1a or Pat1b protein levels to the total protein in each lane using Image Lab (BioRad) and then to the NS DMSO control. Results were plotted in GraphPad and a 2-way ANOVA was performed, ±SEM; n = 3 (Dcp1a) n = 4 (Pat1b), * = P<0.05. D: HeLa Tet-Off cells were transduced with recombinant lentivirus expressing shRNAs targeting NDP52, OPTN, p62 or a NS control and selected with puromycin (1 µg/mL) then cells were co-transfected, treated with Dox and luciferase activity was recorded and analyzed as in Fig 4. Data were plotted in GraphPad as the mean fold change in the relative luciferase activity of each condition compared to vector NS or KapB NS; n = 4. An unpaired t-test was performed; * = P<0.05 ** = P<0.01. E: HUVECs were sequentially transduced first with recombinant lentivirus expressing shNDP52 targeting the 3'-UTR of NDP52 or a NS control and selected with blasticidin (5 µg/mL), and second, with recombinant lentivirus expressing KapB and either mCherry control (mCh) or RFP-NDP52. Coverslips were fixed with 4% paraformaldehyde, permeabilized with 0.1% Triton X-100, and immunostained with Hedls (PBs, green). Scale bar = 20 µm. F: Samples from E were quantified; Hedls puncta were quantified using CellProfiler and presented as number of Hedls puncta per cell, all cells counted are displayed. Results were plotted in GraphPad and a 2-way ANOVA was performed on the main column effects with a Tukey's multiple comparison test, bar represents the mean; n = 3, **** = P<0.0001.

over latency [101]. Naïve iSLKs were infected with WT BAC16 KSHV or delB BAC16 virus for one week to permit latency establishment. Latency was confirmed by immunoblotting for the latent protein LANA which was present in all infected cells to a similar extent, and the lytic protein ORF57 which was absent in the latent iSLK cells one week after infection and present only after reactivation (S7A Fig). LC3-II showed higher accumulation after BafA1 treatment in WT KSHV-infected iSLK cells than those infected with delB BAC16 virus (S7B Fig), confirming that autophagic flux was enhanced in a KapB-dependent manner during KSHV latent infection of iSLK cells. Together, these data validate a critical role for KapB in KSHV enhancement of autophagic flux and show that during this KSHV infection phase, autophagy proceeds via an activation mechanism involving the phosphorylation of Ser90 in Beclin [77].

Previous work by us and others showed that KSHV infection and ectopic expression of KapB activate the stress responsive kinase, MK2 [60,76]. Given our new findings of enhanced autophagic flux by KSHV latency in a KapB-dependent manner, coupled with the prior observation that MK2 promotes starvation-induced autophagic flux via phosphorylation of Beclin, we hypothesized that KapB-mediated MK2 activation may be the mechanism by which autophagic flux is induced our system. We therefore asked the following questions: i) is KapB required for MK2 activation in the context of KSHV latency and ii) is MK2 required for enhanced autophagic flux during KSHV latency? To address these questions, we silenced MK2 in HUVECs prior to infection with WT BAC16 KSHV or delB BAC16 virus for 96 hours to establish latency. Despite trying several different shRNA sequences, our attempts to silence MK2 without significant toxicity to HUVECs were successful only with shRNA1 (Fig 8A and 8B). As a readout for MK2 activation, we immunoblotted for phosphorylated hsp27 (Ser82) [60,76] and showed that KSHV-infected cells displayed increased phosphorylation of hsp27 indicating increased MK2 activation (Fig 8C and 8D). As expected, Ser82 phosphorylation of hsp27 was significantly elevated in latent HUVECs that were infected with WT KSHV, but not in latent HUVECs infected by delB KSHV (Fig 8C); these data support that KapB activates MK2 during KSHV latency. However, when MK2 was silenced, neither WT nor delB latently

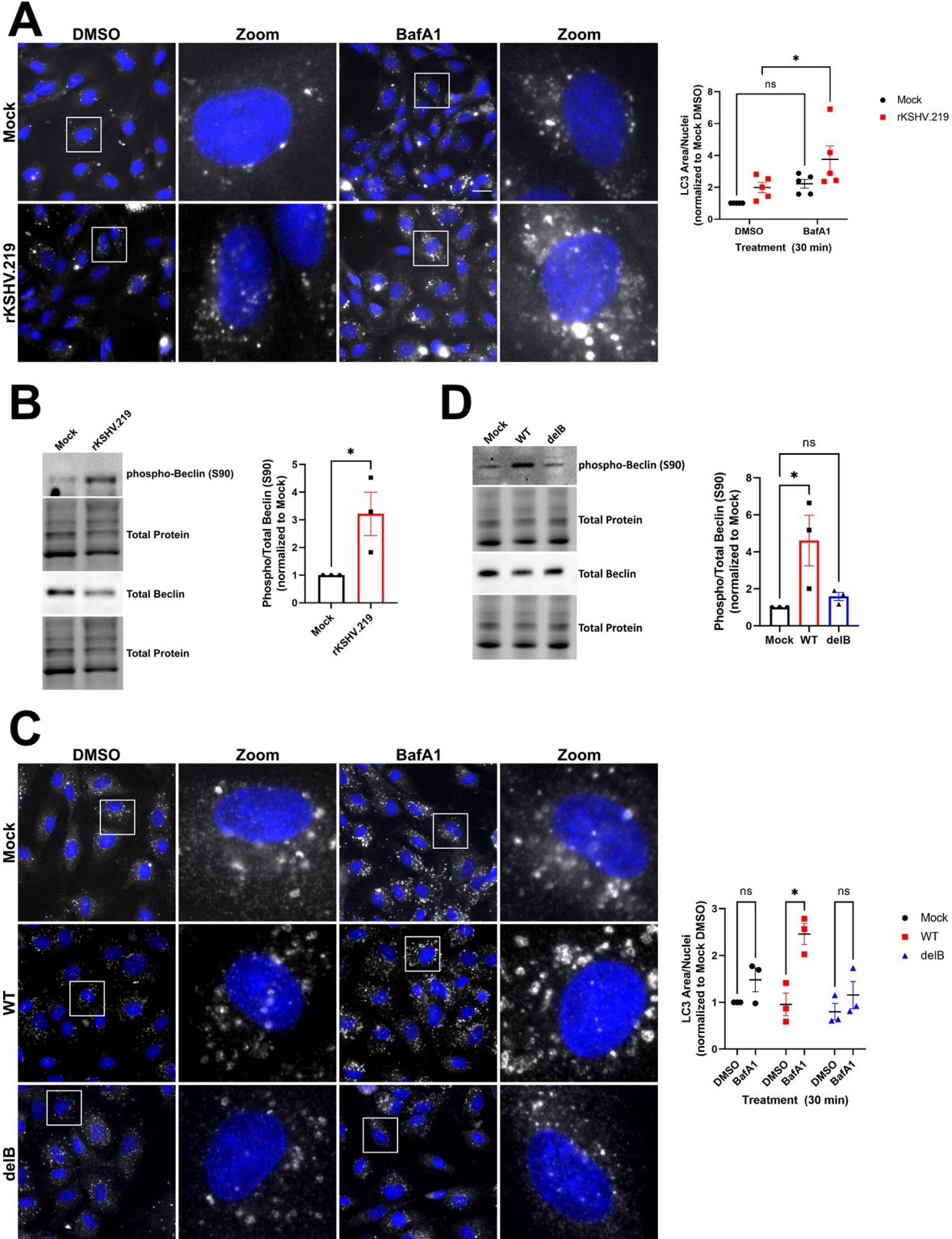

**Fig 7. KSHV infection induces autophagic flux and phosphorylation of Beclin.** A: HUVECs were infected with rKSHV.219 via spinoculation and infection was allowed to proceed for 96 h prior to treatment with Bafilomycin A1 (10 nM) for 30 min. Coverslips were fixed in methanol and immunostained for LC3 (autophagosomes; white), viral latency associated nuclear antigen (LANA, not shown), and DAPI (nuclei, blue). Scale bar = 20 μm. Total LC3 area per field was quantified by identifying LC3 puncta using CellProfiler and normalizing to the number of nuclei and the Mock DMSO control. Results were plotted in GraphPad and a 2-way ANOVA was performed

with a Šidák's multiple comparison test, ±SEM n = 5; * = P<0.05. B: HUVECs were infected as in A for 96 h and harvested in 2X Laemmli buffer. Protein lysates were resolved by SDS-PAGE and immunoblot was performed for phospho-Beclin (S90) and total Beclin. Samples were quantified using Image Lab (BioRad) software and then normalized, first to total protein and then to the Mock control. Results were plotted in GraphPad and a Student's t-test was performed, ±SEM n = 3; * = P<0.05. C: HUVECs were infected with wild-type (WT) or delete KapB (delB) BAC16 KSHV via spinoculation and infection was allowed to proceed for 96 h prior to treatment with Bafilomycin A1 (10 nM) for 30 min. Coverslips were fixed in methanol and immunostained for LC3 (autophagosomes; white), viral LANA (not shown), and DAPI (nuclei, blue). Scale bar = 20 μm. Total LC3 area per field was quantified by identifying LC3 puncta using CellProfiler and normalizing to the number of nuclei and the Mock DMSO control. Results were plotted in GraphPad and a 2-way ANOVA was performed with a Šidák's multiple comparison test, ±SEM n = 3; *P<0.05. D: HUVECs were infected as in C for 96 h and harvested in 2X Laemmli buffer. Protein lysates were resolved by SDS-PAGE and immunoblot was performed for phospho-Beclin (S90) and total Beclin. Samples were quantified using Image Lab (BioRad) software and then normalized, first to total protein and then to the Mock control. Results were plotted in GraphPad and a one-way ANOVA was performed, ±SEM n = 3; * = P<0.05.

infected cells displayed significantly elevated phospho-hsp27 (Fig 8C and 8D). In delB BAC16 infected or uninfected cells, the reduction in phospho-hsp27 in the context of MK2 silencing was not significantly different from control cells whereas WT-infected cells showed a significant reduction in phospho-hsp27 in the context of MK2 silencing. These data confirm that KapB activates MK2 in the context of WT KSHV latency.

To connect MK2 activation to autophagic flux during KSHV latent infection, we silenced MK2 prior to infection and measured autophagic flux by immunoblotting for LC3-II accumulation after BafA1 addition (Fig 8E–8G). LC3-II accumulation was noticeably impaired in both uninfected and KSHV-infected MK2-silenced cells relative to control cells expressing a non-targeting shRNA (Fig 8F), indicating that MK2 contributes to autophagic flux in HUVECs during both uninfected and infected conditions. Because each of these experiments take a significant amount of time, we were concerned that silenced cells adapted to MK2 loss, thereby altering the baseline autophagic flux of silenced cells compared to control cells. To eliminate the bias that this adaptation would have, we directly compared LC3-II accumulation between different infection conditions in MK2-silenced cells (Fig 8G). We compared the following infection conditions: WT-infected vs uninfected, uninfected vs delB infected and WT infected vs delB infected. LC3-II failed to accumulate in WT MK2-silenced cells, and this was significantly different than the rate of accumulation in uninfected MK2-silenced cells (Fig 8G, left graph). This suggests that during KSHV latency, MK2 is a significant contributor to the enhanced autophagic flux observed during infection as in its absence, autophagic flux is impaired. There was no significant difference between LC3-II accumulation in delB infected HUVECs compared to uninfected cells suggesting that during delB KSHV latency, MK2 activation does not contribute to autophagic flux (Fig 8G, middle graph). This is consistent with the failure of delB KSHV infection to promote MK2 activation as evidenced by the lack of enhanced phosphorylation of hsp27 (Fig 8C). Finally, in the context of MK2 silencing, LC3-II accumulation during WT latency was reduced compared to LC3-II accumulation during delB latency, suggesting that KapB activation of MK2 is a major modulator of autophagic flux during KSHV latent infection conditions (Fig 8G, right graph). In summary, MK2 is activated during WT but not delB KSHV latency and using LC3-II accumulation as a measure of autophagic flux, MK2-silenced HUVECs displayed impaired autophagic flux during WT latency specifically when compared to uninfected or delB infected cells. Together, these data confirm that KapB induces autophagic flux during KSHV latency using a mechanism that depends, in large part, on activation of the stress-responsive kinase, MK2.

## Atg5 and NDP52 are required for KSHV-mediated PB disassembly

In our previous work, we showed that KapB expression and KSHV infection induced PB disassembly [60] and more recently we confirmed that KapB was not only sufficient but also

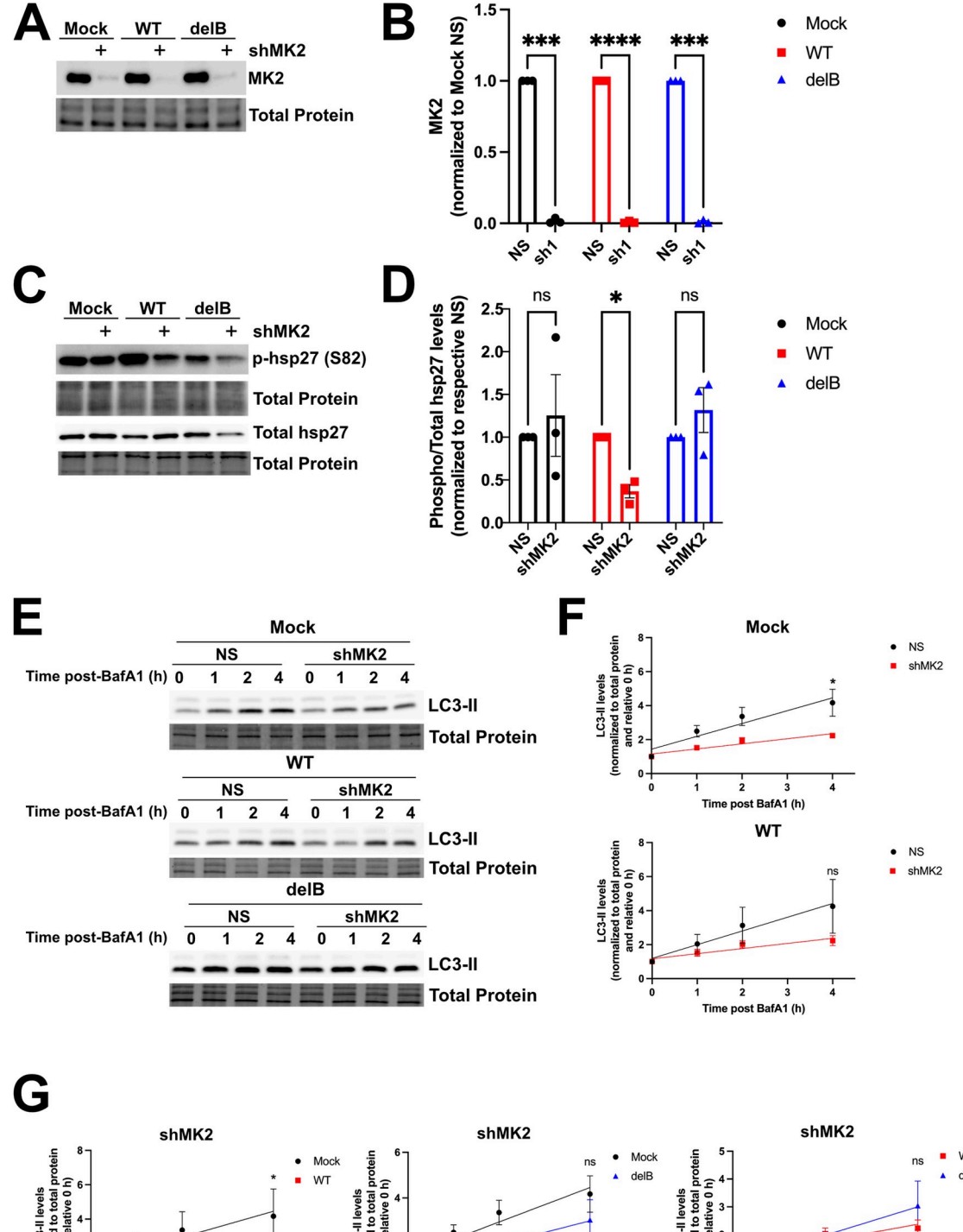

**Fig 8. KSHV-mediated autophagy utilizes MK2.** A: HUVECs were transduced with recombinant lentiviruses expressing shRNAs targeting MK2 or a non-targeting control (NS) and selected with puromycin (1 μg/mL). After selection, transduced cells were infected with wild-type (WT) or delete KapB (delB) BAC16 KSHV via spinoculation and infection was allowed to proceed for 96 h. Cells were harvested in 2X Laemmli buffer. Samples were resolved by SDS-PAGE and immunoblot was performed for MK2. B: Samples from A were quantified for MK2 protein level and normalized to the total protein in each lane using ImageLab software

(BioRad). MK2 levels were then graphed by normalizing to the matched NS for each infection condition in GraphPad, and a one-way repeated measures (RM) ANOVA with a Šidák's multiple comparison test was performed, ±SEM; n = 3, *** = P<0.001, **** = P<0.0001. C: HUVECs were transduced with shMK2-expressing lentiviruses or NS controls, selected, and infected as in A. Cells were harvested in 2X Laemmli buffer. Samples were resolved by SDS-PAGE and immunoblot was performed for total hsp27 and phospho-hsp27 (S82). D: Samples from C were quantified by normalizing phospho- and total hsp27 protein levels to the total protein in each lane using Image Lab (BioRad) then by dividing phospho-hsp27 over total hsp27 and normalizing to the matched NS for mock, WT KSHV and delB infection, respectively. Results were plotted in GraphPad, a one-way RM ANOVA with a Šidák's multiple comparison test was performed, ±SEM; n = 3, * = P<0.05. E: HUVECs were transduced, selected, and infected as in C. Cells were treated with Bafilomycin A1 (BafA1, 10 nM) or a vehicle control (DMSO) for the indicated times prior to harvest in 2X Laemmli buffer. Protein lysates were resolved by SDS-PAGE and immunoblot was performed for LC3. F-G: Samples from E were quantified using Image Lab (BioRad) software and then normalized, first to total protein and then to their respective starting time points (0 h). In F, the accumulation of LC3-II in control cells was compared with accumulation in MK2-silenced cells in either uninfected (top) or WT KSHV infected (bottom) cells. Results were plotted in GraphPad and a linear regression statistical test was performed between control or silenced cells. ±SEM; n = 3, * = P<0.05. In G, results were plotted in GraphPad to show accumulation of LC3-II in MK2-silenced cells to compare between the following groups: WT-infected vs uninfected (left), uninfected vs delB infected (middle) and WT infected vs delB infected (right). A linear regression statistical test was performed between infection conditions in each group as indicated. ±SEM; n = 3, * = P<0.05.

necessary for this effect by using the delB BAC16 virus [102]. To determine if autophagic machinery is required for PB disassembly in the context of KSHV infection, we silenced Atg5 or NDP52 in HUVECs prior to infecting with rKSHV.219 for 96 hours to establish latency, then stained for PBs. Consistent with our data that showed a requirement for autophagy in KapB-mediated PB disassembly, blocking autophagic degradation by silencing Atg5 or NDP52 restored PBs in KSHV-infected cells (Fig 9A and 9B). We quantified PBs and compared their levels between KSHV-infected and mock-infected cells in control cells or in Atg5/NDP52 silenced cells. We did not compare PB counts between KSHV-infected and silenced cells to infected controls because silencing of autophagy regulatory proteins influences PB homeostasis. In control cells that expressed a non-targeting shRNA, KSHV infection decreased PBs, as expected from our previous work (Fig 9A). However, when either Atg5 or NDP52 were silenced, KSHV infection had no effect on PBs relative to the matched uninfected control, and silencing autophagy regulatory proteins restored PBs to levels observed in the absence of infection (Fig 9A). These data confirm an essential role for autophagy and the selective autophagy receptor NDP52 in KSHV-mediated PB disassembly.

To understand more about the molecular mechanism of NDP52-dependent PB disassembly during KSHV infection, we immunoblotted for steady-state levels of NDP52. KSHV infection did not significantly alter the amount of NDP52 protein (Fig 9C). We speculated that rather than influencing NDP52 production, elevated autophagic flux caused by KapB expression, KSHV infection, or Torin enhanced the interaction of NDP52 with its target cargo. As Torin treatment decreased Dcp1a and Pat1b protein levels, and Pat1b was restored by NDP52 silencing (Fig 6C), we tested if either Dcp1a or Pat1b co-immunoprecipitated with NDP52. To determine this, we overexpressed FLAG-tagged NDP52 with GFP-tagged Dcp1a, GFP-tagged Pat1b or a GFP control and immunoprecipitated using a FLAG antibody. We verified immunoprecipitation of NDP52 and that we could detect the expression of GFP and GFP-tagged Dcp1a in whole cell lysate; however, neither GFP alone nor GFP-Dcp1a co-immunoprecipitated with NDP52 (Fig 10A). By contrast, we did co-immunoprecipitate GFP-Pat1b with FLAG-NDP52. These data suggest that NDP52 and Pat1b may interact or reside in the same complex, which could result in delivery of Pat1b and possibly other PB components by NDP52 to the nascent autophagosome for degradation. We reasoned that if NDP52 and Pat1b indirectly interact, KSHV-mediated autophagy should decrease steady-state levels of Pat1b. Consistent with this, Pat1b protein levels were significantly decreased after KSHV infection (Fig 10B). KSHV infection also decreased the steady-state levels of Dcp1a, a result consistent with earlier data that showed Dcp1a decreased in response to KapB expression and Torin

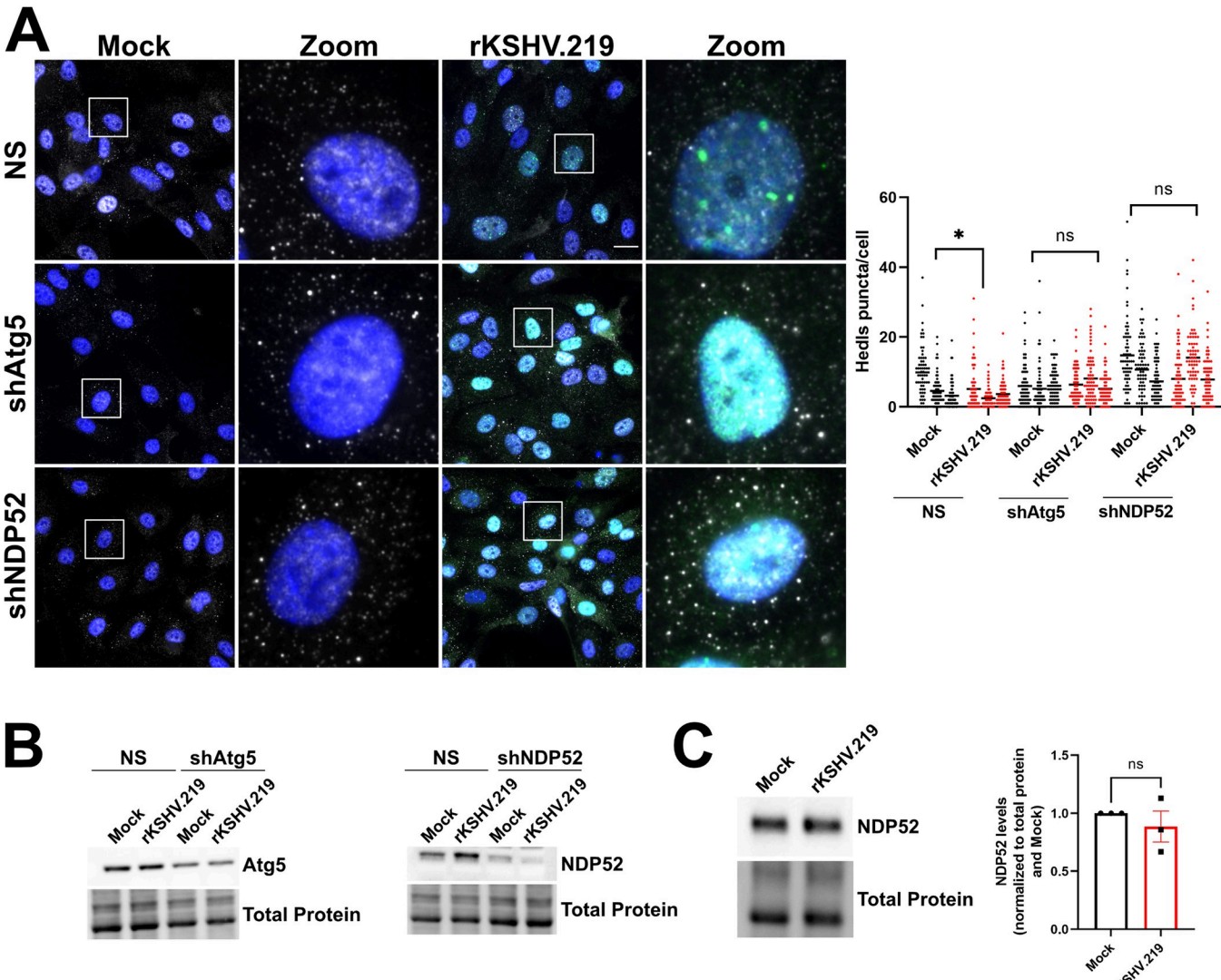

**Fig 9. KSHV-mediated PB disassembly requires Atg5 and the selective autophagy receptor NDP52.** A: HUVECs were transduced with recombinant lentivirus expressing shRNAs targeting Atg5, NDP52, or a non-targeting control (NS) and selected with puromycin (1 μg/mL). Cells were infected with rKSHV.219 via spinoculation and infection was allowed to proceed for 96 h. Coverslips were fixed in 4% paraformaldehyde, permeabilized in 0.1% Triton X-100 and immunostained for Hedls (PBs; white), LANA (KSHV latency, green), and DAPI (nuclei, blue). Scale bar = 20 μm. Hedls puncta were quantified using CellProfiler and presented as number of Hedls puncta per cell, all cells counted are displayed. Results were plotted in GraphPad and a 2-way ANOVA was performed on the main column effects with a Tukey's multiple comparison test, bar represents the mean; n = 3, * = P<0.05. B: HUVECs were transduced with recombinant lentiviruses and infected with rKSHV.219 as in A. Cells were harvested in 2X Laemmli buffer. Protein lysates were resolved by SDS-PAGE and immunoblot was performed for Atg5 and NDP52. C: HUVECs were infected with rKSHV.219 via spinoculation and infection was allowed to proceed for 96 h. Cells were harvested in 2X Laemmli buffer. Samples were resolved by SDS-PAGE and immunoblot was performed for NDP52. Samples were quantified by normalizing NDP52 protein levels to the total protein in each lane using Image Lab (BioRad) and then to Mock. Results were plotted in GraphPad and an unpaired t-test was performed, ±SEM; n = 3, * = P<0.05.

treatment. From these data, we propose the following model: KSHV infection or KapB expression enhances autophagic flux via Beclin phosphorylation while Torin dephosphorylates the ULK complex. In both cases, elevated autophagic flux increases the delivery of Pat1b and some other PB components that interact with Pat1b (such as Dcp1a) to nascent autophagosomes via recognition of Pat1b by the selective autophagy receptor, NDP52. This interaction decreases the abundance of Pat1b, a PB scaffolding protein, resulting in PB disassembly (Fig 10C).

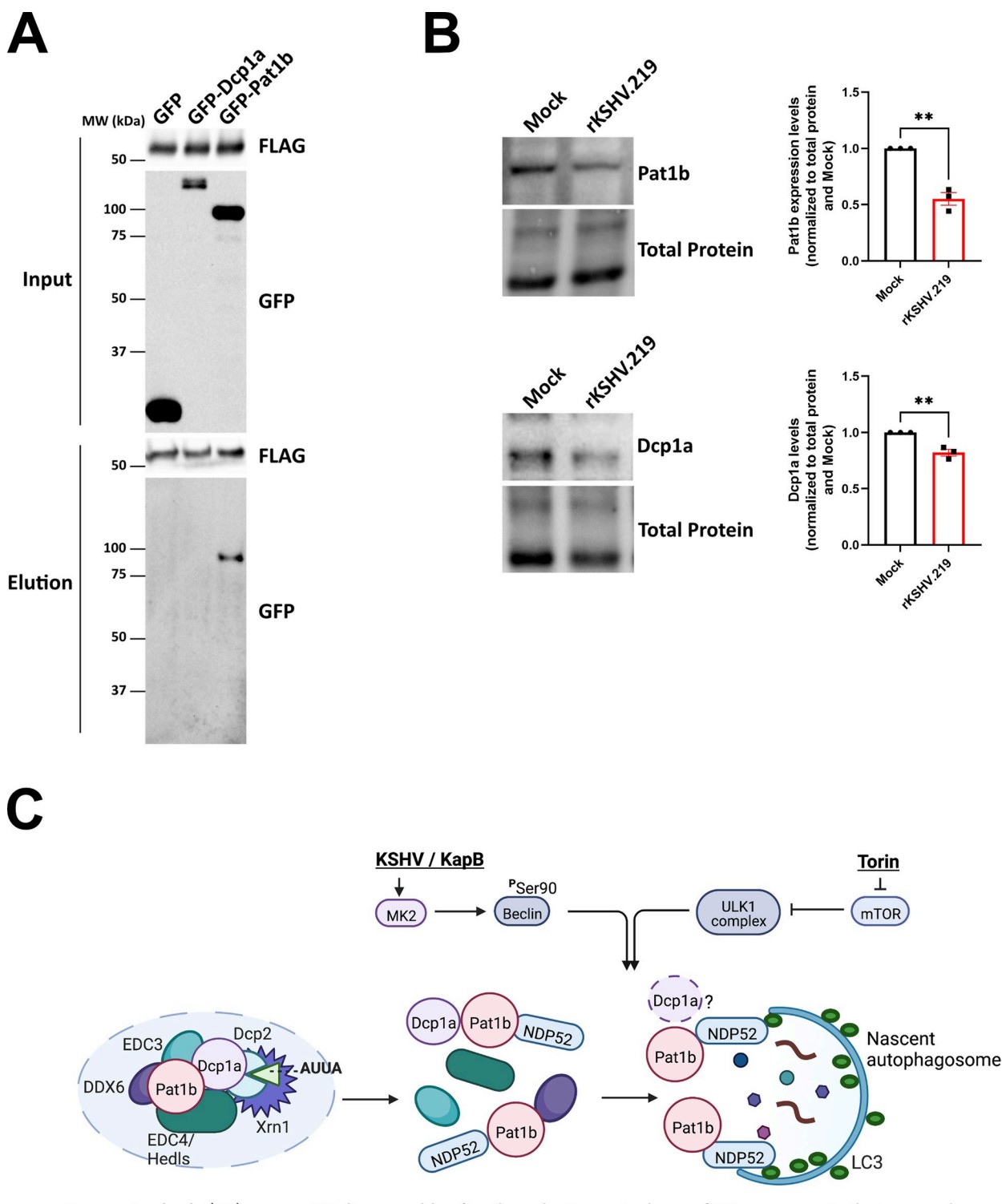

**Fig 10. NDP52 immunoprecipitates with the PB resident protein Pat1b.** A: HEK293T cells were transfected with plasmids expressing pcDNA-FLAG-NDP52 with either pLJM1-GFP (control), pLJM1-GFP-Dcp1a, or pLJM1-GFP-Pat1b for 48 h. Cells were harvested in lysis buffer and incubated with a FLAG mouse antibody (CST) overnight at 4˚C. Immunoprecipitation was performed and samples were eluted from the magnetic beads through boiling in 4X Laemmli buffer (BioRad) with 10% v/v beta-mercaptoethanol. Samples were resolved by SDS-PAGE. A representative blot of three independent experiments is shown. B: HUVECs were infected with rKSHV.219 via spinoculation and infection was

allowed to proceed for 96 h. Cells were harvested in 2X Laemmli buffer. Samples were resolved by SDS-PAGE and immunoblot was performed for Dcp1a or Pat1b. Samples were quantified by normalizing Dcp1a or Pat1b protein levels to the total protein in each lane using Image Lab (BioRad) and then to Mock. Results were plotted in GraphPad and an unpaired t-test was performed, ±SEM; n = 3, ** = P<0.01. C: Model of PB disassembly induced by KSHV/KapB. KapB increases the phosphorylation of Beclin at Serine 90 to enhance autophagic flux during KSHV latency while Torin induces autophagic flux via the inhibition of mTORC1/2 and dephosphorylation of the ULK complex. Enhanced activation of autophagic flux by either mechanism results in PB disassembly that is dependent on the selective autophagy receptor, NDP52. In our model, we predict that NDP52 recruits a PB component(s) to the nascent autophagosome. Our top candidate for this recruitment is the PB scaffolding protein, Pat1b, as NDP52 co-immunoprecipitated with Pat1b and Pat1b protein level decreases after KSHV infection and Torin treatment. Steady-state levels of Dcp1a also decreased in response to KapB expression, KSHV infection, or Torin treatment, suggesting that Dcp1a decreases may be mediated by its partial interaction with Pat1b despite observations that Dcp1a did not co-immunoprecipitate with NDP52. Autophagic degradation of Pat1b and Dcp1a, both key PB scaffolding proteins, leads to PB disassembly. Figure created with Biorender.

## Discussion

Here, we demonstrated that the KSHV protein, KapB, relies on selective autophagy to cause PB disassembly and enhance ARE-mRNA stability. Furthermore, we identified NDP52 as the selective autophagy receptor required for this process. To our knowledge, this is the first report that provides a molecular basis for NDP52-mediated turnover of endogenous PBs and the first description of the contribution of NDP52 to immune regulation by altering inflammatory cytokine RNA turnover/suppression in PBs. We showed that PB disassembly induced by infection with KSHV required Atg5 and NDP52, suggesting an important role for KapB-mediated elevated autophagic flux and PB catabolism in regulating KSHV-associated inflammation. Our major findings are: i) KapB and KSHV latent infection increased autophagic flux by phosphorylation of the autophagy regulatory protein, Beclin, thereby contributing to the complex regulation of autophagic processes during KSHV latent infection and tumourigenesis, ii) KapB expression was necessary for KSHV-mediated MK2 activation and autophagy increases during latent infection, and MK2 was required for KSHV-mediated autophagy increases during latency, iii) KapB- and KSHV-mediated PB disassembly and enhanced translation of an ARE-containing reporter required autophagy machinery and the selective autophagy receptor, NDP52, iv) Activation of autophagy by chemical inhibition of mTORC1/2 using Torin caused PB disassembly using a mechanism that was dependent on autophagic flux and NDP52, suggesting that KapB manipulates an existing cellular pathway that regulates PB catabolism. These results reveal that PB turnover elicited by the KSHV KapB protein during infection requires NDP52 to enhance PB catabolism and promote inflammatory molecule production.

MK2 activation promotes autophagy by phosphorylating Beclin [77]; therefore, we hypothesized that the KSHV protein KapB, a known activator of MK2 [76], would utilize autophagy to promote PB clearance. We tackled this hypothesis using both infection and ectopic expression models. We showed that during latent infection, KSHV elevated Beclin phosphorylation and enhanced autophagic flux as measured by an increase in LC3 puncta size and the accumulation of the lipidated form of LC3-II after Bafilomycin treatment and that LC3-II accumulation during KSHV latency required KapB and MK2. Using ectopic expression, our data showed that both KapB and MK2EE increased autophagic flux via phosphorylation of Beclin at Serine 90 and that KSHV- and KapB-mediated PB disassembly required the autophagy proteins Atg5 and Atg14, and the selective autophagy receptor NDP52. Our data are supported by previous work in yeast where PB recycling required autophagy and in mammalian cells where NDP52 colocalized with an overexpressed GFP-tagged version of the PB protein Dcp1a [100,103]. We add significantly to these studies by performing our work in primary human endothelial cells and by staining for endogenous PBs using immunofluorescence for the decapping co-factor Hedls/EDC4. We revealed that when KapB was expressed, endogenous PBs disassemble via an Atg5-, Atg14-, and NDP52-dependent mechanism. We also complemented

NDP52-silenced cells with an shRNA-resistant NDP52 and observed the restoration of KapB-mediated PB disassembly. This occurred only in cells that expressed KapB and not in control cells that overexpressed NDP52, an observation that suggests that KapB-mediated autophagy increases are required for NDP52-mediated PB disassembly. These data also suggest that NDP52 is required to select the PB protein cargo for autophagic degradation and this is required for KSHV and KapB-mediated PB disassembly, akin to that observed in other studies on NDP52 SAR function [104,105]. Consistent with this, we showed that the PB scaffolding protein, Pat1b, co-immunoprecipitated with NDP52, providing a molecular basis for this interaction. In addition to identifying cargo for clearance, NDP52 has recently been shown to orchestrate autophagosome biogenesis by recruiting essential autophagic machinery to initiation sites [19,20,106] suggesting another possible way that NDP52 may promote KapB-mediated PB disassembly.

Immunoelectron microscopy showed that PBs range in size from 150 to 350 nm and possess an external area of peripheral protrusions that contain the helicase Rck/DDX6 anchored to a dense central core of decay enzymes, including Dcp1a and Xrn1 [46]. Many PB-localized enzymes are required for PB stability including Dcp1a and Pat1b; however, three PB proteins are required for *de novo* PB formation, DDX6, the translation suppressor, 4E-T, and LSM14a [52,99]. Our data showed that KapB expression decreased the steady-state levels of Dcp1a, while the steady-state levels of other PB proteins were unaffected. Inducing autophagy with the mTORC1/2 inhibitor Torin or KSHV infection also decreased steady-state levels of Dcp1a and Pat1b; however only Pat1b protein levels were restored to control levels when NDP52 was silenced in the context of Torin treatment. These data suggest that the entire PB granule is not degraded by autophagy. Rather, our data support a model wherein PB disassembly is induced by the selective degradation of NDP52-associated molecules that likely include Pat1b and possibly the Pat1b-interacting protein, Dcp1a (Fig 10C). Consistent with this, we show that PBs can still form during KapB expression if we overexpress Dcp1a-GFP or block translation using sodium arsenite. Dcp1a is a target for other viruses that mediate PB disassembly; for example, infection with poliovirus caused PB disassembly and the loss of Dcp1a over a time course of infection [107] and adenovirus infection has been shown to re-localize Pat1b from PBs to cytoplasmic aggresomes during infection [108]. We do not yet understand if KSHV infection or Torin treatment alters Pat1b to promote its interaction with NDP52 and subsequent autophagic degradation and PB disassembly; however, we speculate that post-translational modifications of NDP52, Pat1b and perhaps Dcp1a may be involved [98,109–118].

Infection with KSHV induces high levels of pro-inflammatory cytokines, chemokines and angiogenic factors such as TNF, IL-1α, IL-1β, IL-6, CXCL8, IFN-γ, VEGF, COX-2 and GM-CSF [64–70]. Although several KSHV proteins promote the transcription of these mRNAs, many harbor AREs in the 3'UTR of their cognate mRNA, meaning their transcripts are subject to additional post-transcriptional regulation by PBs. We previously showed that KapB and the lytic KSHV protein, viral G-protein-coupled receptor (vGPCR) enhanced the translation of an ARE-containing reporter and elevated steady-state levels of inflammatory products, and that this coincided with the ability of KapB and vGPCR to induce PB disassembly [59,60,74,76]. We now show that KapB expression increased levels of two ARE-containing transcripts, IL-6 and IL-1β, coincident with PB disassembly. However, KapB expression did not consistently increase CXCL8 or COX-2 transcript levels, indicating that PB disassembly is not the sole mechanism of ARE-mRNA decay regulation [119,120]. We also show that the ability of KapB to induce PB disassembly and enhance an ARE-mRNA reporter required both Atg5 and NDP52. The RNA transcripts that encode some autophagy proteins in the Atg5-Atg12-Atg16 complex are also rendered labile by 3'UTR AREs, suggesting that expression of KapB or vGPCR causes a pro-autophagic positive feedback loop where PB disassembly enhances the translation of pro-autophagic products [121,122]. This feature may be aided

further by KapB-mediated upregulation of the ARE-mRNA stabilizing RBP, HuR [123]. This feed forward loop may contribute to the KapB-mediated increase in Atg5 protein levels and the induction of autophagic flux. Taken together, our study reveals a novel way that KSHV infection manipulates autophagic machinery to induce PB turnover and promote inflammatory molecule production. The fact that KSHV encodes more than one protein that reverses the constitutive repression of inflammatory mRNAs in PBs highlights its central importance.

Viral gene products both enhance and block autophagic flux during latent KSHV infection; v-cyclin induces oncogene-induced senescence via induction of autophagy, whereas v-FLIP counteracts v-cyclin, blocking autophagy by binding Atg3 and inhibiting LC3 lipidation [43,44]. We show herein that KapB also upregulates autophagic flux, and that KapB is necessary for the enhanced autophagic flux we observe after KSHV infection of primary endothelial cells. Though these data are consistent with autophagy enhancement observed in latent B-lymphocytes (BCBL-1 cells) [124], they are discordant with a report that KSHV infection dampens autophagic flux and mitophagy [125]. It is not unusual for viruses to display such a push/pull relationship; during coxsackievirus B3 infection, p62 is cleaved by a viral protease to limit its antiviral action while an NDP52 cleavage product retains proviral activity [36]. Such studies lend support to the complex manipulation of selective autophagy pathways emerging for KSHV infection. We considered the possibility that KSHV products may cooperate to promote autophagy during latency via dysregulation of the cellular transcription factor, Nrf2. Nrf2 activity is upregulated by the viral protein vFLIP after *de novo* infection of ECs and during KSHV latency [126,127] and Nrf2 is responsible for the transcriptional activation of several antioxidant response genes as well as NDP52 [128–130]. However, we did not observe KSHV-mediated increases in NDP52 protein after infection, making it more likely that KapB-mediated enhancement of NDP52-dependent selective autophagy of PBs proceeds via post-translational regulation of NDP52 and/or PB components.

In viral cancers like KS, autophagic flux promotes a pro-tumourigenic and pro-inflammatory environment [124,131–133]. The data shown here brings new understanding to this complex relationship during KSHV infection. We show that KapB was both sufficient to induce autophagic flux and that KapB and MK2 were required for enhanced autophagy during KSHV latent infection. We confirmed that autophagy machinery and NDP52 were required for PB disassembly, consistent with our prior work showing that MK2 played an important role in KapB-mediated PB disassembly [60]. Taken together, these data support a requirement for KapB in eliciting the pro-inflammatory environment associated with latent KSHV infection and KS. The manipulation of NDP52 function and autophagic flux by KSHV and KapB is yet another example of the ingenuity with which viruses usurp cellular processes to reveal a novel regulatory network that links NDP52 to inflammatory molecule production via PB catabolism.

## Materials and methods

### Cell culture

All cells were grown at 37˚C with 5% $CO_2$ and 20% $O_2$. HEK293T cells (ATCC), HeLa Tet-Off cells (Clontech), Atg5 +/+ and -/- MEFs [134], HeLa Flp-In TREx GFP-Dcp1a cells (a generous gift from Anne-Claude Gingras), and iSLK.219 cells (a generous gift from Don Ganem) [101] were cultured in DMEM (Thermo Fisher) supplemented with 100 U/mL penicillin, 100 µg/mL streptomycin, 2 mM L-glutamine (Thermo Fisher) and 10% FBS (Thermo Fisher). iSLK.219 cells were additionally cultured in the presence of puromycin (10 µg/mL). HUVECs (Lonza) were cultured in endothelial cell growth medium (EGM-2) (Lonza). HUVECs were seeded onto gelatin (1% w/v in PBS)-coated tissue culture plates or glass coverslips. All drug treatments were performed for the times indicated in each experiment at the concentrations listed in Table 1.

**Table 1. Drugs.**

| Drug | Vendor/Catalogue # | Concentration |
| --- | --- | --- |
| Torin1 | Sigma-Aldrich 475991 | 250 nM |
| Bafilomycin A1 | Sigma-Aldrich B1793 | 10 nM |
| Puromycin | ThermoFisher A1113803 | 1 μg/mL (HUVECs) or 10 μg/mL (iSLKs) |
| Blasticidin | ThermoFisher A1113903 | 5 μg/mL |
| Polybrene | Sigma-Aldrich H9268 | 5 μg/mL |
| Sodium Arsenite | Sigma-Aldrich 1.06277 | 0.25 mM |
| Doxycycline | Sigma-Aldrich D9891 | 1 μg/mL |
| Hygromycin B | ThermoFisher 10687010 | 500 or 1200 μg/mL |

## Cloning

All plasmids used in this study can be found in Table 2. All shRNAs were generated by cloning shRNA hairpin sequences found in Table 3 into pLKO.1-TRC Puro, pLKO.1-TRC cloning vector was a gift from David Root (Addgene plasmid # 10878; http://n2t.net/addgene:10878; RRID: Addgene_10878), or pLKO.1-blast, pLKO.1-blast was a gift from Keith Mostov (Addgene plasmid #26655; http://n2t.net/addgene:26655; RRID:Addgene_26655). pcDNA-FLAG-NDP52 was made by amplifying NDP52 from RFP-NDP52 (a generous gift from Dr. Andreas Till) using

**Table 2. Plasmids.**

| Plasmid | Use | Source | Mammalian Selection |
| --- | --- | --- | --- |
| RFP-NDP52 | Cloning | Dr. Andreas Till (University Hospital of Bonn) | N/A |
| pcDNA 3.1 (+) | Empty Vector Control | Invitrogen V79020 | N/A |
| FLAG-HA-pcDNA | Empty Vector Control Co-IP | Addgene #52535 [140] | N/A |
| pcDNA KapB BCBL1 | ARE-mRNA stability | [60] | N/A |
| pcDNA FLAG-MK2EE | ARE-mRNA stability | [60] | N/A |
| pLKO.1-TRC | shRNA expression | Addgene #10878 [141] | Puromycin |
| pLKO.1-blast | shRNA expression | Addgene #26655 [142] | Blasticidin |
| pLJM1 | Empty Vector Control | [143] | Blasticidin/ Puromycin |
| pLJM1 KapB (pulmonary KS) | Overexpression | Cloned from pBMNIP-KapB [60] into pLJM1-BSD | Blasticidin |
| pLJM1 mCh | Overexpression Control | Craig McCormick (Dalhousie University) | Blasticidin |
| pLJM1 FLAG-MK2EE | Overexpression | Cloned into pLJM1 using BamHI and EcoRI from pBMN FLAG-MK2EE [60] | Blasticidin/ Puromycin |
| pLJM1 RFP-NDP52 | Overexpression | Cloned into pLJM1 using NheI and BamHI from RFP-NDP52 | Puromycin |
| pcDNA-FLAG-NDP52 | Co-IP | Cloned using primers listed in Table 4 from RFP-NDP52 | N/A |
| pLJM1-EGFP | Co-IP | Addgene #19319 [144] | Puromycin |
| pT7-EGFP-C1-HsDcp1a | Cloning | Addgene #25030 [145] | N/A |
| pLJM1-GFP-Dcp1a | Co-IP | Cloned into pLJM1 using AgeI and EcoRI from pT7-EGFP-C1-HsDcp1a | Puromycin |
| Pat1b-GFP | Cloning | [146] | N/A |
| pLJM1-GFP-Pat1b | Co-IP | Cloned into pLJM1 using NheI and BamHI from GFP-Pat1b | Puromycin |
| pMD2.G | Lentivirus generation | Addgene #12259 | N/A |
| psPAX2 | Lentivirus generation | Addgene #12260 | N/A |
| pTRE2 Firefly Luciferase-ARE | ARE-mRNA stability | [74] | N/A |
| pTRE2-Renilla Luciferase | ARE-mRNA stability | [74] | N/A |

**Table 3. shRNA target sequences.**

| shRNA target | shRNA sequence 5'→3' | Region |
|---|---|---|
| Non-targeting (NS) | AGCACAAGCTGGAGTACAACTA | |
| Atg14 | GTCTGGCAAATCTTCGACGAT | CDS |
| Atg5 | CTTTGATAATGAACAGTGAGA | CDS |
| MK2 | AGAAAGAGAAGCATCCGAAAT | CDS |
| NDP52-1 | GAGCTGCTTCAACTGAAAGAA | CDS |
| NDP52-2 | GACTTGCCTATGGAAACCCAT | CDS |
| NDP52-3 | CCCTTTGTGAACTAAGTTCAA | 3'-UTR |
| NDP52-4 | CCTGACTTGATACTAAGTGAT | 3'-UTR |
| Optineurin-1 | GCACGGCATTGTCTAAATATA | CDS |
| Optineurin-2 | GCCATGAAGCTAAATAATCAA | CDS |
| p62 | CCGAATCTACATTAAAGAGAA | CDS |
| NBR1-1 | GCCAGGAACCAAGTTTATCAA | CDS |
| NBR1-2 | CCATCCTACAATATCTGTGAA | CDS |
| VCP-1 | ACCGTCCCAATCGGTTAATTG | CDS |
| VCP-2 | AGATCCGTCGAGATCACTTTG | CDS |

the primers found in Table 4 and performing a restriction digest with EcoRI and BamHI to insert into FLAG-HA-pCDNA, FLAG-HA-pcDNA3.1 was a gift from Adam Antebi (Addgene plasmid # 52535; http://n2t.net/addgene:52535; RRID:Addgene_52535). pLJM1-GFP-Dcp1a was generated from pT7-EGFP-C1-HsDcp1a, pT7-EGFP-C1-HsDCP1a was a gift from Elisa Izaurralde (Addgene plasmid # 25030; http://n2t.net/addgene:25030; RRID:Addgene_25030) through restriction digest with AgeI and EcoRI to insert into pLJM1. pLJM1-GFP-Pat1b was generated from Pat1b-GFP (a generous gift from Dr. Nancy Standart) through restriction digest with NheI and BamHI to insert into pLJM1.

## Lentivirus production and transduction

All lentiviruses were generated using a second-generation system. Briefly, HEK293T cells were transfected with psPAX2, pMD2G, and the plasmid containing a gene of interest or hairpin using polyethylimine (PEI, Polysciences). psPAX2 was a gift from Didier Trono (Addgene plasmid #12260; http://n2t.net/addgene:12260; RRID:Addgene_12260) and pMD2.G was a gift from Didier Trono (Addgene plasmid #12259; http://n2t.net/addgene:12259; RRID: Addgene_12259).

Viral supernatants were harvested 48 h post-transfection, clarified using a 0.45 μm poly-ethersulfone (PES) filter (VWR), and frozen at -80°C until use. For transduction, lentiviruses were thawed at 37°C and added to target cells in complete media containing 5 μg/mL poly-brene (Sigma) for 24 h. The media was changed to selection media containing 1 μg/mL puro-mycin or 5 μg/mL blasticidin (Thermo) and cells were selected for 48 h before proceeding with experiments.

**Table 4. Cloning Primers.**

| Primer | Primer sequence |
|---|---|
| NDP52-EcoRI-Forward | 5' ACCGGTCCATGGAGGAGACCATCAAA 3' |
| NDP52-BamHI-Reverse | 5' CCGGTGGATCCTCAGAGAGA 3' |

## Viral conditioned media treatments

iSLK.219 cells (latently infected with rKSHV.219 virus) were sub-cultured into a 10 cm cell culture dish and grown without puromycin for 72 h. At 72 h, conditioned media was reserved and cleared of cellular debris by centrifugation at 500 x *g* for 5 min. Supernatant was collected and filtered through a 0.22 μM PES membrane filter (VWR) and stored in aliquots at -80˚C. Prior to the experiment, conditioned media was thawed at 37˚C and combined 1:1 with fresh HUVEC EGM-2 media and used to treat vector or KapB-expressing HUVECs for 0 or 6 h prior to total RNA harvest.

## Production of KSHV

Cloning and creation of the iSLK-BAC16 delKapB cell line is described in [102]. iSLK.219 or iSLK-BAC16 cells were sub-cultured into a 10 cm cell culture dish without puromycin and reactivated from latency with either 1 μg/mL doxycycline (Sigma-Aldrich) alone (iSLK.219) and 1 μg/mL doxycycline and 1 mM sodium butyrate (Sigma-Aldrich) (iSLK-BAC16). After 72 h, supernatants were collected, centrifuged to remove cellular debris, aliquoted, and stored at -80˚C until use.

## KSHV infection

Sub-confluent (50–70%) 12-well or 6-well plates of HUVECs or naïve iSLKs [101], respectively, were incubated with either rKSHV.219 or KSHV BAC16 viral inoculum diluted in antibiotic and serum free DMEM containing 5 μg/mL polybrene. Plates were centrifuged at 800 x *g* for 2 h at room temperature [135], the inoculum was removed and replaced with either EGM-2 or antibiotic free 10% FBS DMEM. For KSHV BAC16 infections, WT and delKapB inoculums were normalized empirically prior to infection to circumvent our previously observed impairment in latency establishment during delKapB infection [102]. To establish robust and stable latency in Fig 7, *de novo* infected iSLKs were expanded from a 6-well to a 10cm dish, selected with 500 μg/mL hygromycin B (ThermoFisher), split once, then selected with 1200 μg/mL hygromycin B for a combined period of one week prior to seeding. Where indicated, latent iSLKs were induced to reactivate using 1 μg/mL doxycycline for 48 h.

## Immunofluorescence

Cells were seeded onto coverslips for immunofluorescence experiments and fixed for 10 min at 37˚C in 4% (v/v) paraformaldehyde (Electron Microscopy Sciences). Samples were permeabilized with 0.1% (v/v) Triton X-100 (Sigma-Aldrich) for 10 min at room temperature and blocked in 1% human AB serum (Sigma-Aldrich) for 1 h at room temperature. For LC3 immunofluorescence cells were fixed for 10 min in ice cold methanol at -20˚C and blocked in 1% human AB serum for 1 h at room temperature. Primary and secondary antibodies were diluted in 1% human AB serum and used at the concentrations in Table 5. Samples were mounted with Prolong Gold AntiFade mounting media (Thermo). Image analysis was performed using CellProfiler (cellprofiler.org), an open source platform for image analysis [136]; quantification of puncta was performed as previously described, with the exception that cells were defined by propagating out a set number of pixels from each nucleus [137].

## Co-Immunoprecipitation

HEK293T cells were transfected with plasmids expressing either FLAG-HA-pCDNA (empty vector) or pcDNA-FLAG-NDP52 with either pLJM1-GFP (control), pLJM1-GFP-Dcp1a, or pLJM1-GFP-Pat1b for 48 h. Cells were harvested in lysis buffer (150 mM NaCl, 10 mM Tris pH 7.4, 1 mM EDTA, 1% v/v Triton X-100, 0.5% v/v NP-40, Roche protease inhibitor tablet)

**Table 5. Antibodies.**

| Antibody | Species | Vendor/Catalogue # | Application | Dilution |
|---|---|---|---|---|
| Hedls | Mouse | Santa Cruz sc-8418 | Immunofluorescence<br>Immunoblot | 1:1000<br>1:1000 |
| KapB | Rabbit | A generous gift from Don Ganem | Immunofluorescence<br>Immunoblot | 1:1000<br>1:1000 |
| NDP52 | Rabbit | CST 60732 | Immunoblot | 1:1000 |
| p62 | Rabbit | CST 7695 | Immunoblot | 1:1000 |
| Dcp1a | Mouse | CST 15365 | Immunoblot | 1:500 |
| DDX6 | Rabbit | Bethyl A300-461 | Immunoblot | 1:1000 |
| Pat1b | Rabbit | CST 14288 | Immunoblot | 1:500 |
| LC3B | Rabbit | CST 2775 | Immunofluorescence<br>Immunoblot | 1:200<br>1:1000 |
| Atg5 | Rabbit | CST 2630 | Immunoblot | 1:1000 |
| Atg14 | Rabbit | CST 96752 | Immunoblot | 1:1000 |
| OPTN | Rabbit | Abcam ab23666 | Immunoblot | 1:1000 |
| Xrn1 | Rabbit | Abcam ab231197 | Immunoblot | 1:1000 |
| VCP | Rabbit | CST 2649 | Immunoblot | 1:1000 |
| NBR1 | Rabbit | CST 9891 | Immunoblot | 1:1000 |
| LANA | Rabbit | A generous gift from Don Ganem | Immunofluorescence<br>Immunoblot | 1:1000<br>1:1000 |
| Phospho-S6 (Ser235/236) | Rabbit | CST 4858 | Immunoblot | 1:1000 |
| Total S6 | Rabbit | CST 2217 | Immunoblot | 1:1000 |
| Phospho-Beclin 1 (Ser90) | Rabbit | CST 86455 | Immunoblot | 1:500 |
| Total Beclin 1 | Mouse | CST 4122 | Immunoblot | 1:1000 |
| Phospho-ULK (Ser555) | Rabbit | CST 5869 | Immunoblot | 1:1000 |
| Phospho-ULK (Ser757) | Rabbit | CST 14202 | Immunoblot | 1:1000 |
| Total ULK | Rabbit | CST 8054 | Immunoblot | 1:1000 |
| FLAG | Rabbit | CST 14793 | Immunoblot | 1:1000 |
| FLAG | Mouse | CST 8146 | Immunoprecipitation | 1:100 |
| GFP | Rabbit | CST 2956 | Immunoblot | 1:1000 |
| MK2 | Rabbit | CST 12155S | Immunoblot | 1:1000 |
| Total hsp27 | Mouse | CST 2401S | Immunoblot | 1:1000 |
| Phospho-hsp27 (Ser82) | Rabbit | CST 2401S | Immunoblot | 1:1000 |
| ORF57 | Mouse | Santa Cruz sc-135746 | Immunoblot | 1:1000 |

and incubated with a FLAG mouse antibody (CST) overnight at 4˚C. Protein G magnetic beads (BioRad) were incubated overnight at 4˚C with 5 mg/mL BSA in lysis buffer without Triton X-100 or NP-40. Immunoprecipitation was performed and samples were eluted from the magnetic beads through boiling in 4X Laemmli buffer (BioRad) with 10% v/v beta-mercaptoethanol.

## Immunoblotting and densiometric analysis of steady state protein levels

Cells were lysed in 2X Laemmli buffer and stored at -20˚C until use. The DC Protein Assay (Bio-Rad) was used to quantify protein concentration as per the manufacturer's instructions. 10–15 µg of protein lysate was resolved by SDS-PAGE on TGX Stain-Free acrylamide gels (BioRad). Total protein images were acquired from the PVDF membranes after transfer on the ChemiDoc Touch Imaging system (BioRad). Membranes were blocked in 5% bovine serum albumin (BSA) or skim milk in Tris-buffered saline-Tween20 (TBS-T). Primary and secondary

**Table 6. qPCR primers.**

| Primer | Primer Sequence |
| --- | --- |
| HPRT-1- Forward | 5' CTTTCCTTGGTCAGGCAGTATAA 3' |
| HPRT-1—Reverse | 5' AGTCTGGCTTATATCCAACACTTC 3' |
| 18S - Forward | 5' TTCGAACGTCTGCCCTATCAA 3' |
| 18S - Reverse | 5' GATGTGGTAGCCGTTTCTCAGG 3' |
| IL-6—Forward | 5' GAAGCTCTATCTCGCCTCCA 3' |
| IL-6—Reverse | 5' TTTTCTGCCAGTGCCTCTTT 3' |
| CXCL8—Forward | 5' AAATCTGGCAACCCTAGTCTG 3' |
| CXCL8—Reverse | 5' GTGAGGTAAGATGGTGGCTAAT 3' |
| IL-1β - Forward | 5' CTCTCACCTCTCCTACTCACTT 3' |
| IL-1β - Reverse | 5' TCAGAATGTGGGAGCGAATG 3' |
| COX-2—Forward | 5' CCCTTGGGTGTCAAAGGTAA 3' |
| COX-2—Reverse | 5' GCCCTCGCTTATGATCTGTC 3' |

antibodies were diluted in 2.5% BSA or skim milk, dilutions can be found in Table 5. Membranes were visualized using Clarity Western ECL substrate and the ChemiDoc Touch Imaging system (BioRad).

Densiometric analysis of protein bands following immunoblotting was performed using stain-free gel and blot technology using ImageLab software (BioRad). Briefly, a trihalo compound is present in the acrylamide solution that covalently binds to tryptophan residues within the electrophoresed protein when the acrylamide gel is activated with UV light (Chemi Doc Touch, BioRad). Activation of the trihalo compound adds 58 Da moieties to available tryptophan residues which permits detection of total protein transferred to the PVDF membrane with a sensitivity comparable to Coomassie Blue. After images of total protein were obtained, standard immunoblotting and densiometric quantification was performed, ensuring that band intensities were in the linear range for both the protein of interest and the total protein according to published methods [138,139]. In all statistical analysis of densitometry results, we quantified bands from three (or more) independent experiments.

## Luciferase assays

Luciferase assays were performed as previously described [74]. HeLa Tet-Off cells were transduced with recombinant lentivirus expressing different shRNAs and selected. Cells were transfected according to [74] with pTRE2 Firefly Luciferase ARE (FLuc), pTRE2 Renilla Luciferase (RLuc), and the expression plasmid of interest using Fugene HD (Promega). FLuc and RLuc activity were quantified using the Dual Luciferase Assay Kit (Promega) and read on a GloMax multi-detection system (Promega).

## Quantitative PCR

RNA was collected using a RNeasy Plus Mini Kit (Qiagen) according to the manufacturer's instructions and stored at -80°C until further use. RNA concentration was determined and was reverse transcribed using Maxima H Minus First Strand cDNA Synthesis Kit (Thermo-Fisher) using a combination of random and oligo (dT) primers according to the manufacturer's instructions. cDNA was diluted either 1:5 or 1:10 for all qPCR experiments and GoTaq qPCR Master Mix (Promega) was used to amplify cDNA. The ΔΔquantitation cycle (Cq) method was used to determine the fold change in expression of target transcripts. qPCR primer sequences can be found in Table 6.

## Statistics

Data shown are the mean ± standard error of the mean (SEM). Statistical significance was determined using a one-way a 2-way ANOVA (with the applicable post-test indicated in the corresponding figure legend), a Student's t-test or a simple linear regression analysis to compare differences between slopes (permits comparison between two best fit lines not comparison between multiple samples in one statistical test). All statistics were performed using GraphPad Prism v.9.0.

## Supporting information

**S1 Fig. KapB expression promotes PB disassembly but does not prevent PB assembly.**
HeLa cells expressing a doxycycline-inducible GFP-Dcp1a were used to determine whether KapB expression prevented granule assembly or promoted disassembly. A: HeLa cells were transfected with either KapB or an empty vector control prior to inducing expression of GFP-Dcp1a with doxycycline (1 μg/mL). Cells were fixed 12 h post-induction and immunostained for Hedls (PBs, red) and KapB (blue). Scale bar = 20 μm. B: GFP-Dcp1a expression was induced with doxycycline (1 μg/mL) in HeLa cells prior to transfection with either KapB or an empty vector control. Cells were fixed 12 h post-transfection and immunostained for Hedls (PBs, red) and KapB (blue). Scale bar = 20 μm.
(TIFF)

**S2 Fig. Loss of Atg5 prevents KapB-mediated PB disassembly.** A&B: HUVECs were sequentially transduced: first with recombinant lentiviruses expressing either shRNAs targeting Atg5 or Atg14 (shAtg5, shAtg14) or a non-targeting control (NS) and selected with puromycin (1 μg/mL), and second with either KapB or an empty vector control and selected with blasticidin (5 μg/mL). Samples were lysed in 2X Laemmli and resolved by SDS-PAGE. Samples were immunoblotted for Atg5 (A) or Atg14 (B). For Atg5 (A), representative blot is from the same membrane, hashed line indicates skipped lanes. For Atg5 (A) samples were quantified by normalizing Atg5 protein levels to the total protein in each lane using Image Lab (BioRad) and then to Vector NS. Results were plotted in GraphPad, a 2-way ANOVA was performed, ±SEM; n = 3, *** = P<0.001. C: Atg5 +/+ (WT) and -/- (KO) MEFs were transduced as in A and harvested in 2X Laemmli buffer before being resolved by SDS-PAGE. Samples were probed for LC3, a representative blot is shown. D: Atg5 +/+ and -/- MEFs were transduced with recombinant viruses expressing KapB or an empty vector control and selected with blasticidin (5 μg/mL). Samples were fixed in 4% paraformaldehyde and permeabilized in 0.1% Triton X-100. Immunofluorescence was performed for Hedls (PBs, white) and DAPI (nuclei, blue). Scale bar = 20 μm. Hedls puncta were quantified using CellProfiler and presented as number of Hedls puncta per cell, all cells counted are displayed. Results were plotted in GraphPad and a 2-way ANOVA was performed on the main column effects with a Tukey's multiple comparison test, bar represents the mean; n = 3, ** = P<0.01.
(TIFF)

**S3 Fig. Autophagy gene silencing in HeLa cells.** HeLa cells were transduced with recombinant lentiviruses expressing either shRNAs targeting Atg5 or Atg14 (shAtg5, shAtg14) or a non-targeting control (NS) and selected with puromycin (1 μg/mL) prior to transfection with KapB or an empty vector for luciferase assays. Samples were lysed in 2X Laemmli buffer and resolved by SDS-PAGE before immunoblotting with Atg5 (A) or Atg14 (B). Representative blots are shown.
(TIFF)

**S4 Fig. Selective autophagy receptor knockdown in HUVECs reveals that OPTN and p62 are not required for KapB-mediated PB disassembly.** A&B: HUVECs were transduced with shRNAs targeting VCP, NBR1, or a non-targeting control (NS) and selected with puromycin (1 µg/mL), then transduced with an empty vector control or KapB and selected with blasticidin (5 µg/mL). Samples were lysed in 2X Laemmli buffer, resolved by SDS-PAGE, and immuno-blotted for NBR1 (A) and VCP (B). Representative blots are shown. Successful VCP knock-down was lethal while NBR1 knockdown was not successful. C: HUVECs were transduced with shRNAs targeting NDP52 or a non-targeting control (NS) and selected with puromycin (1 µg/mL), then transduced with an empty vector control or KapB and selected with blasticidin (5 µg/mL). Samples were lysed in 2X Laemmli buffer, resolved by SDS-PAGE, and immuno-blotted for NDP52. D&E: HUVECs were transduced with shRNAs targeting p62, OPTN, or a non-targeting control (NS) and selected with puromycin (1 µg/mL), then transduced with an empty vector control or KapB and selected with blasticidin (5 µg/mL). Samples were lysed in 2X Laemmli buffer and resolved by SDS-PAGE, and immunoblotted for p62 (D) or OPTN (E). Representative blots are shown. F: HUVECs were transduced as in D and E. Coverslips were fixed in 4% paraformaldehyde, permeabilized in 0.1% Triton X-100 and immunostained for Hedls (PBs; white), DAPI (nuclei, blue). Scale bar = 20 µm. Hedls puncta were quantified using CellProfiler and presented as number of Hedls puncta per cell, all cells counted are displayed. Results were plotted in GraphPad and a 2-way ANOVA was performed on the main column effects with a Tukey's multiple comparison test, bar represents the mean; n = 3, ** = P<0.01. G: HUVECs were transduced with recombinant lentiviruses targeting NDP52 or a non-targeting control (NS) and selected with puromycin (1 µg/mL). Samples were lysed in 2X Laemmli buffer, resolved by SDS-PAGE, and immunoblotted for NDP52 to confirm knock-down for Torin immunofluorescence experiments performed in parallel.
(TIFF)

**S5 Fig. shRNA silencing of selective autophagy receptors in HeLa cells.** HeLas were trans-duced with shRNAs targeting NDP52, OPTN, p62, or a non-targeting control (NS) and selected with puromycin (1 µg/mL) prior to transfection with KapB or an empty vector for luciferase assays. Samples were lysed in 2X Laemmli buffer, resolved by SDS-PAGE, and immunoblotted for NDP52 (A), p62 (B), and OPTN (C). Representative blots are shown.
(TIFF)

**S6 Fig. NDP52 shRNA silencing and rescue in control HUVECs.** A: Representative western blot of HUVECs transduced with an shRNA targeting the 3'UTR of NDP52 or a non-targeting control (NS) and selected with puromycin (1 µg/mL), then co-transduced with an empty vec-tor control and mCherry (mCh) or KapB and mCh. Samples were lysed in 2X Laemmli, resolved by SDS-PAGE, and immunoblotted for NDP52. B: HUVECs were sequentially trans-duced first with recombinant lentivirus expressing shNDP52 targeting the 3'-UTR of NDP52 or a NS control and selected with blasticidin (5 µg/mL), and second, with vector and either an mCherry control (mCh) or RFP-NDP52. Coverslips were fixed with 4% paraformaldehyde, permeabilized with 0.1% Triton X-100, and were immunostained with Hedls (PBs, green). Scale bar = 20 µm. C: Hedls puncta were quantified using CellProfiler and presented as num-ber of Hedls puncta per cell, all cells counted are displayed. Results were plotted in GraphPad and a 2-way ANOVA was performed on the main column effects with a Tukey's multiple com-parison test, bar represents the mean; n = 3.
(TIFF)

**S7 Fig. KapB is required for KSHV-induced autophagic flux in latent iSLK cells.** A-B: Naïve iSLK cells were infected with wild-type (WT) or delete KapB (delB) BAC16 KSHV via

spinoculation, robust and stable latency was achieved by expanding and selecting cells in hygromycin B (500 μg/mL and then 1200 μg/mL) for one week before seeding for experiments. A: After one week, cells were either lysed for immunoblotting or treated with 1 μg/mL Dox to induce reactivation (WT only) and lysed for immunoblotting 48 h after reactivation. One representative immunoblot of three independent experiments is shown. B: Cells were treated with Bafilomycin A1 (BafA1, 10 nM) or a vehicle control (DMSO) for the indicated times prior to harvest in 2X Laemmli buffer. Protein lysates were resolved by SDS-PAGE and immunoblot was performed for p62 and LC3. Samples were quantified using Image Lab (BioRad) software and then normalized, first to total protein and then to their respective starting time points (0 h). Results were plotted in GraphPad and a linear regression statistical test was performed, ±SEM; n = 3, * = P<0.05.
(TIFF)

## Acknowledgments

The authors would like to dedicate this manuscript to Dr. Beth Levine, who was a leader in the field of autophagy and viruses and who inspired this work. We sincerely thank the members of the Corcoran lab for helpful discussions. We would like to thank Dr. Craig McCormick (Dalhousie University), Dr. Eric Pringle (Dalhousie University), and Dr. Andrew Leidal (University of Calgary) for valuable feedback on this manuscript and the McCormick lab for plasmids, expertise, and advice. We would like to thank Mr. Stephen Whitefield and Ms. Mary Ann Trevors in the Dalhousie *CORES* (Cellular and Molecular Digital Imaging and Electron Microscopy core facilities) and Dr. Anne Vaahtokari of the Charbonneau Microscopy Facility, UCalgary for microscopy support. We would like to thank Dr. Andreas Till (University of Bonn), Dr. Nancy Standart (University of Cambridge), Dr. Anne-Claude Gingras (Lunenfeld-Tanenbaum Research Institute) and Dr. Don Ganem (Novartis) for sharing valuable reagents. Model figure was created with Biorender.

## Author Contributions

**Conceptualization:** Carolyn-Ann Robinson, Gillian K. Singh, Jennifer A. Corcoran.

**Formal analysis:** Carolyn-Ann Robinson, Gillian K. Singh, Mariel Kleer, Thalia Katsademas, Elizabeth L. Castle.

**Funding acquisition:** Jennifer A. Corcoran.

**Investigation:** Carolyn-Ann Robinson, Gillian K. Singh, Mariel Kleer, Thalia Katsademas, Bre Q. Boudreau, Jennifer A. Corcoran.

**Methodology:** Carolyn-Ann Robinson, Gillian K. Singh, Mariel Kleer, Elizabeth L. Castle, Bre Q. Boudreau.

**Project administration:** Jennifer A. Corcoran.

**Resources:** Jennifer A. Corcoran.

**Software:** Elizabeth L. Castle.

**Supervision:** Jennifer A. Corcoran.

**Writing – original draft:** Carolyn-Ann Robinson, Gillian K. Singh, Jennifer A. Corcoran.

**Writing – review & editing:** Carolyn-Ann Robinson, Gillian K. Singh, Mariel Kleer, Thalia Katsademas, Bre Q. Boudreau, Jennifer A. Corcoran.

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
