## [Decision Letter · Decision Letter 0]

22 Feb 2022

Dear Dr. Corcoran,

Thank you very much for submitting your manuscript "Kaposi’s sarcoma-associated herpesvirus (KSHV) utilizes the NDP52/CALCOCO2 selective autophagy receptor to disassemble processing bodies" for consideration at PLOS Pathogens. As with all papers reviewed by the journal, your manuscript was reviewed by members of the editorial board and by several independent reviewers. In light of the reviews (below this email), we would like to invite the resubmission of a significantly-revised version that takes into account the reviewers' comments.

We cannot make any decision about publication until we have seen the revised manuscript and your response to the reviewers' comments. Your revised manuscript is also likely to be sent to reviewers for further evaluation.

Sincerely,

Fanxiu Zhu, Ph.D.

Associate Editor

PLOS Pathogens

Shou-Jiang Gao

Section Editor

PLOS Pathogens

Kasturi Haldar

Editor-in-Chief

PLOS Pathogens

orcid.org/0000-0001-5065-158X

Michael Malim

Editor-in-Chief

PLOS Pathogens

orcid.org/0000-0002-7699-2064

Reviewer's Responses to Questions

**Part I - Summary**

Reviewer #1: This is a revised version of the manuscript that focuses on KSHV KapB modulation of NDP52-mediated selective autophagy to target PBs for disassembly. In previous version, the key issue was the biological relevance of these findings to KSHV infection. In the revision, the authors added new results (Fig 6) to address this issue.

Overall, a lot of experiments have been performed in this project. The involvement of NDP52-mediated selective autophagy in KSHV latency is interesting and novel topic. Most experiments were well controlled and rigorously designed with valid rationale. The data showing that the autophagy plays a role in PB loss (as indicated by Dcp1a loss) is solid, as evidenced by different approaches, including Torin1 inhibition of mTOR, BafA1 inhibition of autophagy activity, and ATG5 depletion.

Reviewer #2: The authors have responded point-by-point to the reviewers’ comments. The use of recombinant KSHV viruses, including delB BAC16, makes the paper stronger. Nonetheless, there remain still some concerns to address.

Reviewer #3: In this revised manuscript (PPATHOGENS-D-22-00182), the authors explored the interplay between KapB and processing bodies. They show that KapB directly influences Beclin phosphorylation leading to autophagic pathways activation and further PB disassembly. The authors have a long history of studying this process, and in this article they provide important new details on the mechanism underlying KapB-mediated PB disassembly. This is a technically challenging mechanism to study and a complex multiplayer story. Yet, the authors do an elegant balancing act of explaining their results while providing sufficient details to support their claims. Based on the previous round of reviews, there were concerns about physiological relevance of the findings and the authors have greatly added to their original manuscript to address these concerns. In particular, they have generated a KapB deletant KSHV backbone to test directly the effect of KapB on this cascade and have infected primary cells to show that the observed effect on autophagic flux was KapB-dependent. The findings are important and of interest for the KSHV community.

**Part II – Major Issues: Key Experiments Required for Acceptance**

Reviewer #1: 1. The biological relevance issue has been addressed with de novo KSHV infection assays. The use of a KapB-deficient KSHV is a powerful tool in this regard (Fig 6, C-D). However, the KapB-deficient KSHV was only used for primary infection, instead of latent infection (which was done for KSHV.219, Fig 6A). This project focuses on KSHV latency, and it is notable that autophagy plays different or even opposite roles in primary vs latent viral infection. Furthermore, the outcomes were only assessed by confocal microscopy for LC puncta deregulation, which is not convincing. More important evaluations including immunoblotting for PB loss (Dcp1a as a marker) and autophagy involvement should be done to consolidate the conclusion and improve the consistency of data throughout the project.

2. Another important experiment to consolidate the biological relevance is to deplete KapB (by shRNA or sgRNA) in KSHV HUVECs, followed by evaluation of relevant outcomes and phenotypes. I appreciate they authors did loss-of-function assays using shRNA-mediated knockdown for NDP52 and ATG5, but similar experiments for KapB are more important in terms of biological relevance.

3. It would be also important to check if KSHV latent infection induces NDP52 expression, considering the fact that EBV induces the selective autophagy receptor p62 in its latency and employs p62-mediated selective autophagy for DNA damage response.

4. The link of MK2 to KapB-elicited autophagy is missing.  That is said, the authors’ hypothesis that KapB induces autophagy through a MK2-dependent manner was not tested in the manuscript, although the authors have shown in their previous papers that KapB interacts with MK2, and MK2 can phosphorylate Beclin1. A loss-of-function assay of MK2 or an assay with a KapB MK2-interacting deficient mutant is needed to answer this question.

Reviewer #2: 1. While the authors provided the biochemical data on how KapB induces autophagy flux, how KapB activates selectively NDP52-mediated autophagy is not clear yet. Fig. 7C depicts the direct interaction between Dcp1a (or other PB protein) and NDP52, but the manuscript does not show any biochemical evidence to support their interactions.

2. The other SAR Calcoco1 and Calcoco3/TAX1BP1 are paralogs that most resemble NDP52. Thus, it’d be nice to comparatively examine the activities of Calcoco1 and Calcoco3/TAX1BP1 in PB dismantling, Dcp1a interaction, and ARE-mRNA reporter expression along with NDP52.

3. The authors determined the relative intensities of immunoblotting (IB) bands only by normalizing to that of control band. However, some IB samples were not equally loaded cross the lanes, shown by the levels of total proteins. Equal loading of the samples, particularly in Figs. 1A, 2E, 4B, 4D, 7B (shAtg5), S2A, S2B, S3A, S3B, and S5A, may not support the conclusions drawn in this paper.

Reviewer #3: N/A

**Part III – Minor Issues: Editorial and Data Presentation Modifications**

Reviewer #1: Other issues (most are actually not minor):

1. Since the central finding of this study is that NDP52-mediated selective autophagy is involved in PB disassembly in KSHV latency, one outstanding question is about the definition of NDP52-mediated selective autophagy. In terms of p62-mediated selective autophagy, it requires substantial post-translational modification on p62, i.e. site-specific phosphorylation and ubiquitination, which are required for target recognition and transportation. The authors should evaluate site-specific phosphorylation (and/or ubiquitination) of NDP52, which is likely required for its function in PB clearance, in the context.

2. Fig 1A. It seems like the basal autophagy activity in HUVECs is very high, compared 0 h between Vector vs KapB samples. LC3-II should be at very low level if the basal autophagy activity is low. The high basal level of autophagy activity in HUVECs is further shown by low level of p62 and PB components such as Dcp1a (Fig 4, B, D). Thus, further induction of autophagy by overexpressed KSHV KapB in these cells may not be significant. This is particularly true considering the marginal difference of p62 downregulation (the endogenous level of p62 is already very low due to high endogenous autophagy activity) in the Vector vs KapB group in immunoblotting results (Fig 1A).

3. Fig 4D. The results are not convincing and the current experimental design for disclosing role of Atg5 is not correct. The current design is to disclose KapB role in ATG5-mediated upregulation of Dcp1a. On the figure, I do not see KapB causes a loss of Dcp1a (lanes 1 vs 3), nor Atg5 depletion restores any loss of Dcp1a (lanes 2 vs 4). The statistical analysis on lanes 3 vs 4 is not correct since this comparison cannot disclose a role for Atg5 in this process.

4. Evidence on Dcp1a loss caused by autophagy induction (e.g. Torin1 inhibition) is solid. However, Recapitulation with KapB on Dcp1a loss is not very convincing. For example, Fig 4B. The IB blot on KapB downregulation of steady-state Dcp1a (also p62) is not convincing; no difference between lanes 1 vs 3 on the blot. The authors should show the best IB result from the triple independent experiments.

5. Fig 2. Page 12. To my knowledge, ULK1 S555 is targeted by AMPK for phosphorylation, not a target of mTOR.

6. Fig 5. Immunoblotting is needed to evaluate Dcp1a recovery of Torin1-induced loss in NDP52-depleted HUVECs, although different approaches, including loss-of-function and rescue assays were used. Immunoblotting on Dcp1a regulation would make the data consistent in the study and consolidate the conclusion.

7. It is an interesting question if NDP52-mediated autophagy for PB clearance also happens in KSHV PEL cells. This study only used HUVECs. This can be discussed.

8. Fig2B. Statistical analysis is missing

Reviewer #2: 4. Like p62 in Fig. 1A, is it expected that the level of NDP52 is decreased by KapB but restored by Baf A1 treatment?

5. The data of the NDP52 complementation experiments in Figs. 5D/5E and S6B/S6C seem controversial. In Figs. S6B and S6C, why could not NDP52 complementation in shNDP52 control cells induce PB disassembly?

6. In Fig. 3A, does KapB expression significantly enhance the expression of IL-6 at 0 h? If not, like Torin (Fig. 3B), is it expected that the iSLK 219 culture media can promote IL-6 expression in the presence of Torin?

7. Otherwise, it’d be better to remove the results of the experiments using the conditional media in Fig. 3A. It is not clear if the media promoted the expression of the target genes tested, as not provided the statistical significance between 0 and 6 h in either vector or KapB cells.

If the authors anticipate a synergistic or additive role of the conditional media and KapB in the expression of ARE-containing cytokine genes, it is possible also to consider that proinflammatory cytokines induced in KapB-overexpressed cells or the context of latently infected (KapB+) cells may promote PB autophagy, i.e, via autocrine signaling in addition to KapB-mediated MK2 activation. This possibility may be examined using the conditional media generated from wild-type and delB BAC16 iSLK cells.

Reviewer #3: - Primary KSHV infection induces autophagy despite vFLIP: have the author check vFLIP expression in these cells?

- I understand that they author tried to mimic natural infection by infecting HUVECS but why not use cells where KSHV latency is established like BCBL or iSLKs where they could directly compare the effect to the lytic cycle?

- Figure S1 is important – consider adding it as a main figure

- No the authors’ fault but the quality of the figures I received was bad (very pixelized) so please ensure that figure quality is checked

PLOS authors have the option to publish the peer review history of their article (what does this mean?). If published, this will include your full peer review and any attached files.

Reviewer #1: No

Reviewer #2: No

Reviewer #3: No
---

## [Decision Letter · Decision Letter 1]

7 May 2022

Dear Dr. Corcoran,

Thank you very much for submitting your manuscript "Kaposi’s sarcoma-associated herpesvirus (KSHV) utilizes the NDP52/CALCOCO2 selective autophagy receptor to disassemble processing bodies" for consideration at PLOS Pathogens. As with all papers reviewed by the journal, your manuscript was reviewed by members of the editorial board and by several independent reviewers. Two reviewers are still not satisfied by the revision. In light of the reviews (below this email), we would like to invite the resubmission of a significantly-revised version that takes into account the reviewers' comments. You may address their concerns by either additional experiments or sound arguments.

We cannot make any decision about publication until we have seen the revised manuscript and your response to the reviewers' comments. Your revised manuscript is also likely to be sent to reviewers for further evaluation.

Sincerely,

Fanxiu Zhu, Ph.D.

Associate Editor

PLOS Pathogens

Shou-Jiang Gao

Section Editor

PLOS Pathogens

Kasturi Haldar

Editor-in-Chief

PLOS Pathogens

orcid.org/0000-0001-5065-158X

Michael Malim

Editor-in-Chief

PLOS Pathogens

orcid.org/0000-0002-7699-2064

The reviewers still have some concerns. Please address those with your best effort by either additional experiments or sound arguments.

Reviewer's Responses to Questions

**Part I - Summary**

Reviewer #1: The revised manuscript has been somewhat improved, with some new data added that address some of previous critiques. However, most of the key critiques have not been sufficiently addressed. My impression remains that evidence on Dcp1a loss caused by autophagy induction (e.g. Torin1 inhibition) is convincing, but recapitulation with KSHV KapB on Dcp1a loss is not solid. I am sorry I do not feel comfortable if I recommend to accept the current version.

Reviewer #2: The manuscript has been revised but the authors have not addressed the major and minor points raised by the reviewer in an appropriate manner. The new results are not convincing, and there are unsolved statistical issues and inappropriate data analysis and interpretation.

**Part II – Major Issues: Key Experiments Required for Acceptance**

Reviewer #1: 1. The involvement of MK2 in KapB-induced autophagy. The proposed pathway “KapBMK2Becline1 phosphorylationautophagy” is not complete since evidence for the involvement of MK2 is not provided. As Reviewer 2 also pointed out in previous version, “how KapB activates selectively NDP52-mediated autophagy is not clear yet”. The authors argued that they had difficulties in silencing MK2 in HUVECs. However, they did silence ATG4 and ATG14 (Fig 2), and ATG5 and NDP52 (Fig 8) in these cells. So there is no technical challenges. Replacement of different shRNAs should address this issue. Also, as suggested, a KapB deletion mutant that lacks MK2-interacting ability can somewhat alternatively address this issue since the authors successfully transfected KapB into HUVECs.

2. Loss-of-function assay in already established KSHV latency. The authors argued that the use of KapB deficient KSHV virus for infection is a far superior approach. I agreed in previous version this truncated virus is a powerful tool. However, as Reviewer 3 also pointed, “…..why not use cells where KSHV latency is established like BCBL or iSLKs where they could directly compare the effect to the lytic cycle?”. First, the authors did not show evidence that latency is established 96h postinfection. Second, I still think performing loss-of-function assays in already established KSHV latency are necessary to consolidate the conclusion. As shown above, the authors have no technique challenges to silence a gene in KSHV latency.

Reviewer #2: 1. According to the reviewer’s comment, the authors performed a co-immunoprecipitation assay to examine whether NDP52 interacts with Dcp1a and the other PB protein Pat1b. However, the authors found that Dcp1a was not readily detected in the NDP52-immunoprecipitate (IP) complex. This is an unexpected result and different from the model proposed in the original manuscript, Instead, the authors found that Pat1b could be co-precipitated as shown in Fig. 8D. Unfortunately, the data was not appropriately presented; the author should show a full image of the GFP blot of Elution along with a molecular weight marker to verify that the control GFP alone was not co-precipitated with Flag-NDP52 (lane 1).

If Dcp1a is expected to bind indirectly to NDP52 via Part1b as depicted in Fig. 8F, it should be readily detected in the NDP52-IP complex precipitated from the lysates of cells co-transfected with GFP-Pat1b.

Furthermore, the authors did not provide evidence yet to support “direct” interaction between NDP52 and Pat1b, for example, using a GST pull-down assay with purified proteins.

2. The authors added new data to Fig. 6C, showing that torin-induced degradation of Pat1b was inhibited by NDP52 silencing. However, the degradation of its partner Dcp1a was not inhibited by NDP52 silencing. This is not in line with the model proposed by the authors that NDP52 and Pat1b could serve to deliver Dcp1a to the autophagosomes for degradation.

3. Recently, Calcoco1 has been known to bind to LC3 and serve as selective autophagy receptor (SAR) for the ER-phagy (Nthiga et. al., (2020) EMBO J. 39:e103649) and golgiphagy (Nthiga et. al., (2021) Autophagy 17:2051). In addition, TAX1BP1/Calcoco3 is known to act as SAR for xenophagy and aggrephagy. So, it was highly interesting to examine the activities of the NDP52 relatives in PB disassembly in addition to (or rather than) p62 and OPTN as performed in Fig. S4. This reviewer believes that it is important to examine if there is a functional redundancy for PB autophagy among the Calcoco family proteins.

4. The paper used the unusual quantification method to normalize and determine the relative intensities of immunoblot bands rather than using a loading control blot, such as Actin or GAPDH. It is still confusing. The authors did not mention a MW range of total protein blots used in normalization. In addition, it is very strange that the graphs of the relative band intensities were not changed even after the normalization by total protein blots. That is, the new graphs are exactly the same as the old ones. In addition, the authors did not provide the relative band intensities in some IB data. If normalized by total in Figs. 8B and S5A, the levels of Atg5 are unlikely reduced by shAtg5.

The authors did not describe how they performed the immunoblots for statistical analyses in Figs. 1A, 3E, 5B, 6C, 7B, 7D, 7E, 8C, and 8E. Where were they from different blots of the same lysates or each blot from three to four independent experiments?

5. The comments raised by the reviewer in Fig. 3A (new Fig. 4A) were simply 1) to provide the statistical evaluation (either ns or p-value) of the expression of IL-6 between control and KapB cells at 0 h, 2) to statistically identify the effects of the conditional media in either control or KapB cells on cytokine expression, and 3) to examine the combined effects of the conditional media and torin in cytokine expression in Fig. 3B.

The authors mentioned that KapB expression and transcriptional stimulus (the conditional media) significantly enhanced IL-6 and IL-1 beta levels. However, it is unlikely that IL-1 beta expression in KapB cells is enhanced by addition of the media for 6 h, that is, no effect of the conditional media.

**Part III – Minor Issues: Editorial and Data Presentation Modifications**

Reviewer #1: 3. Fig 4A (new Fig 5A). Again, the current data show that silencing ATG5 upregulates Dcp1a, but it is not correct that silencing ATG5 restores KapB-mediated Dcp1a loss.

4. “If basal autophagy is low, there should be equal levels of LC3-I and LC3-II in the cell”. This statement is incorrect. LC3-II is an autophagic cleavage product of LC3-I. If the basal level of autophagy is low, LC3-II level should be low (could be undetectable by IB). So, again, in Fig 1A, if KapB induces autophagy, we should see elevated LC3-II level at 0 h in KapB vs Control samples (Labe 1 vs Lane 3), and LC3-II should be low at 0h in Control sample (lane 1) if the basal level of autophagy in HUVECs is low.

5. “We tried to immunoblot for vFLIP to detect its expression using two different commercial antibodies, without success”. This is an unacceptable excuse. KSHV vFLIP has been well studied and IB results are presented in tones of papers.

6. Since the authors failed to show KSHV induces NDP52, which is expected in that KSHV activates NRF2 that induces NDP52, the possibilities should be discussed.

New critiques:

7. Canonical vs noncanonical autophagy. The authors make inconsistent claims on canonical vs noncanonical autophagy forms induced by KapB. For example, “KapB failed to activate autophagic flux by canonical pathways that are used by Torin”. “Canonical autophagy is required for KapB-mediated increases in ARE-containing cytokine transcripts”. Please clarify what is the exact form of autophagy and what are their differences.

8. Line 159: Subtitle: KapB-mediated PB disassembly requires Atg5 and Atg14. Line 322: Subtitle: Atg5 and NDP52 are required for KSHV-mediated PB disassembly. Reorganization is needed to put ATG5 data together.

Reviewer #2: (No Response)

PLOS authors have the option to publish the peer review history of their article (what does this mean?). If published, this will include your full peer review and any attached files.

Reviewer #1: No

Reviewer #2: No
---

## [Decision Letter · Decision Letter 2]

22 Nov 2022

Dear Dr. Corcoran,

Thank you very much for submitting your manuscript "Kaposi’s sarcoma-associated herpesvirus (KSHV) utilizes the NDP52/CALCOCO2 selective autophagy receptor to disassemble processing bodies" for consideration at PLOS Pathogens. As with all papers reviewed by the journal, your manuscript was reviewed by members of the editorial board and by several independent reviewers. The reviewers appreciated the attention to an important topic. Based on the reviews, we are likely to accept this manuscript for publication, providing that you modify the manuscript according to the review recommendations.

Sincerely,

Fanxiu Zhu, Ph.D.

Academic Editor

PLOS Pathogens

Shou-Jiang Gao

Section Editor

PLOS Pathogens

Kasturi Haldar

Editor-in-Chief

PLOS Pathogens

orcid.org/0000-0001-5065-158X

Michael Malim

Editor-in-Chief

PLOS Pathogens

orcid.org/0000-0002-7699-2064

Reviewer Comments (if any, and for reference):

Reviewer's Responses to Questions

**Part I - Summary**

Reviewer #1: I have no more significant concerns on the revision.

Reviewer #2: The authors have successfully addressed the previous concerns, except #3 and #5, raised by this reviewer.

**Part II – Major Issues: Key Experiments Required for Acceptance**

Reviewer #1: None

Reviewer #2: None

**Part III – Minor Issues: Editorial and Data Presentation Modifications**

Reviewer #1: Still think these two sections should be re-organized regarding ATG5. It is confusing in the current format

1. KapB-mediated PB disassembly requires Atg5 and Atg14

2. Atg5 and NDP52 are required for KSHV-mediated PB disassembly

Reviewer #2: The #3 concern was whether the other NDP52 paralogs, Calcoco1 and Calcoco3, are involved in PB disassembly. This reviewer thought that the experiment would be straightforward when the authors have the cell lines that are knocked down or out for the genes. The new data could be added to Fig. S4 along with the data of p62 and OPTN. Nonetheless, I understand and agree to the authors’ response.

The concerns in #5 were 1) to provide the statistical evaluation (either ns or p-value) of the expression of IL-6 between control and KapB cells at 0 h, 2) to statistically identify the effects of the conditional media in either control or KapB cells on cytokine expression, and 3) to examine the combined effects of the conditional media and torin in cytokine expression in Fig. 4B. The authors have not responded to the second and third concerns. The last concern was about the relationship between transcription and autophagy regulation of the expression of the cytokine genes. And the following new text is not easy for me to follow. It’d be nice to rewrite.

“We then examined levels of some endogenous ARE-containing RNA transcripts in KapB expressing cells (Fig 4A). After KapB expression, IL-1b mRNA was significantly elevated

compared to equivalently treated controls both before and after conditioned media treatment;

however, the IL-6 was transcript was significantly increased only after conditioned media (Fig

4A). We interpret these data as follows. First, transcriptional stimulus did not further enhance IL-

1b mRNA levels, as the steady-state of this mRNA was already elevated in the context of KapB

expression. Second, KapB-mediated elevation of IL-6 transcript could be detected only after

transcriptional stimulus, suggesting that in the absence of transcriptional upregulation IL-6

mRNA levels were not sufficient to benefit from the absence of PB turnover. Together, these

data suggest that some cytokine transcripts respond to KapB-mediated PB disassembly with

elevated levels either at baseline or upon transcriptional induction that we suggest is a result of

reduced transcript decay in PBs (Fig 4A). In addition, treatment with Torin promoted the

enhanced steady-state levels of IL-1β and COX-2 (Fig 4B), suggesting that inducing autophagy

and PB disassembly by an alternative mechanism also reduced the decay of some ARE-mRNA

transcripts. This suggests that cytokine mRNAs that shuttle to PBs are likely to respond to

diverse stimuli that upregulate autophagic flux.”

PLOS authors have the option to publish the peer review history of their article (what does this mean?). If published, this will include your full peer review and any attached files.

Reviewer #1: No

Reviewer #2: No

Figure Files:

Data Requirements:

Reproducibility:

References:

---

## [Editor Report · Decision Letter 3]

22 Dec 2022

Dear Dr. Corcoran,

We are pleased to inform you that your manuscript 'Kaposi’s sarcoma-associated herpesvirus (KSHV) utilizes the NDP52/CALCOCO2 selective autophagy receptor to disassemble processing bodies' has been provisionally accepted for publication in PLOS Pathogens.

Best regards,

Fanxiu Zhu, Ph.D.

Academic Editor

PLOS Pathogens

Shou-Jiang Gao

Section Editor

PLOS Pathogens

Kasturi Haldar

Editor-in-Chief

PLOS Pathogens

orcid.org/0000-0001-5065-158X

Michael Malim

Editor-in-Chief

PLOS Pathogens

orcid.org/0000-0002-7699-2064
---

## [Editor Report · Acceptance letter]

9 Jan 2023

Dear Dr. Corcoran,

We are delighted to inform you that your manuscript, "Kaposi’s sarcoma-associated herpesvirus (KSHV) utilizes the NDP52/CALCOCO2 selective autophagy receptor to disassemble processing bodies," has been formally accepted for publication in PLOS Pathogens.

Best regards,

Kasturi Haldar

Editor-in-Chief

PLOS Pathogens

orcid.org/0000-0001-5065-158X

Michael Malim

Editor-in-Chief

PLOS Pathogens

orcid.org/0000-0002-7699-2064